# Geometry-aware RL for Manipulation of Varying Shapes and Deformable Objects

**Tai Hoang**[1][*]**, Huy Le**[1,2]**, Philipp Becker**[1]**, Ngo Anh Vien**[2]**, Gerhard Neumann**[1]
[1]Autonomous Learning Robots, Karlsruhe Institute of Technology
[2]Bosch Center for Artificial Intelligence

## Abstract

Manipulating objects with varying geometries and deformable objects is a major challenge in robotics. Tasks such as insertion with different objects or cloth hanging require precise control and effective modelling of complex dynamics. In this work, we frame this problem through the lens of a heterogeneous graph that comprises smaller sub-graphs, such as actuators and objects, accompanied by different edge types describing their interactions. This graph representation serves as a unified structure for both rigid and deformable objects tasks, and can be extended further to tasks comprising multiple actuators. To evaluate this setup, we present a novel and challenging reinforcement learning benchmark, including rigid insertion of diverse objects, as well as rope and cloth manipulation with multiple end-effectors. These tasks present a large search space, as both the initial and target configurations are uniformly sampled in 3D space. To address this issue, we propose a novel graph-based policy model, dubbed *Heterogeneous Equivariant Policy (HEPi)*, utilizing $SE(3)$ equivariant message passing networks as the main backbone to exploit the geometric symmetry. In addition, by modeling explicit heterogeneity, HEPi can outperform Transformer-based and non-heterogeneous equivariant policies in terms of average returns, sample efficiency, and generalization to unseen objects. Our project page is available here.

## 1 Introduction

Geometric structure plays a crucial role in robotic manipulation. For instance, in insertion tasks, a robot must precisely align objects with their corresponding target placements. Understanding the geometries is therefore essential in such tasks, as each pair requires a unique alignment (Zeng et al., 2020; Tang et al., 2024). Similarly, for deformable objects like cloths, whose shapes change over time, successfully completing the task requires a policy that can capture these dynamic geometric changes (Lin et al., 2021; Antonova et al., 2021; Shi et al., 2024). In both scenarios, these geometric structures can be naturally represented as graphs, a widely adopted framework in robot learning (Wang et al., 2018; Huang et al., 2020; Ryu et al., 2023; Shi et al., 2024). In this paper, we frame manipulation problems as heterogeneous graphs. Taking the *Cloth-Hanging* task as an example, depicted in Figure 1, the cloth and the actuators are represented as two distinct node sets, connected by a set of directed inter-edges. Each node is associated with a geometric vector representing its 3D coordinates. Yet, this representation results in high-dimensional observation and action spaces, which makes learning policies that generalize seamlessly to novel orientations, poses, and unseen geometries challenging.

To address this issue, recent works (Zeng et al., 2020; Huang et al., 2022; 2024; Ryu et al., 2023) introduced equivariance in the $SE(3)$ space as an inductive bias. Using Equivariant Message Passing Networks (EMPNs), they learn policies that generalize to different poses by leveraging the geometric structure of the scene. These works learn by imitation, and most only consider simple pick-and-place tasks, where the model only has to produce a desired end-effector pose to be reached by a controller. An exception is the recent Equibot (Yang et al., 2024), where the policy outputs velocity vectors rather than static end-effector poses, enabling success in more complex and dynamic tasks like cloth folding and wrapping by imitation. This work investigates how to transfer these ideas from imitation

---

[*]Correspondence to tai.hoang@kit.edu

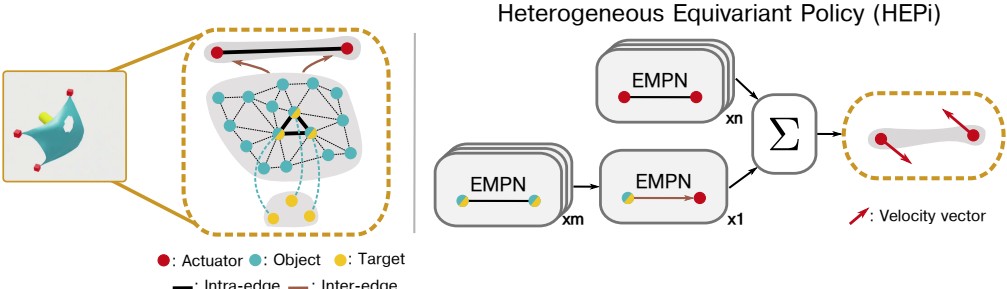

Figure 1: **Left:** A *Cloth-Hanging* task represented by a heterogeneous graph that comprises two disjoint node sets, objects, and actuators, connected through directed, fully-connected inter-edges. Intra-edges occur within each set (both objects and actuators) to capture relationships within clusters. Information is aggregated from objects to actuators via inter-edges. The target distance is absorbed into the feature representation rather than treated as a separate node type. **Right:** Overview of *Heterogeneous Equivariant Policy (HEPi)*, consisting of multiple Equivariant Message Passing Networks (EMPNs) process the graph, and the outputs are aggregated to generate the final action.

to reinforcement learning. Unlike supervised imitation, training policies with reinforcement learning presents additional challenges, particularly due to the need for high-frequency data collection and efficient adaptation to new experiences. Large policy networks struggle in these settings, as they are unable to quickly adapt to changing data (Andrychowicz et al., 2021). To address these issues, we design a lightweight heterogenous equivariant architecture, amenable to efficient on-policy reinforcement learning. The architecture's equivariance allows generalizing between poses and its heterogeneity enables us to include and exploit knowledge about the scene as well as the unactuated and actuated objects in it. For training, we find that naively using Proximal Policy Optimization (PPO) (Schulman et al., 2017), can result in suboptimal performance, and we propose to employ a more principled trust region approach from Otto et al. (2021) to achieve stable convergence.

To evaluate our approach and future advancements in this direction, we propose a novel suite of seven tasks, realized using NIVIDA IsaacLab (Mittal et al., 2023) to utilize its GPU-based simulation engine. They are designed to highlight the role of geometric structure in manipulation tasks, with a progressive increase in difficulty, from simple rigid-body manipulation with diverse objects to more challenging tasks involving multiple actuators and deformable objects. Our experimental results demonstrate that the proposed *Heterogeneous Equivariant Policy (HEPi)* outperforms both Transformer-based and pure EMPN baselines, particularly in complex 3D manipulation tasks. HEPi's integration of equivariance and explicit heterogeneity modelling improves performance in terms of average returns, sample efficiency, and generalization to unseen objects.

To summarize, our contributions are **i)** a novel benchmark comprising rigid insertion of varying geometries and deformable objects manipulation that is particularly well-suited for geometry aware reinforcement learning research; **ii)** HEPi, a graph-based policy that is expressive and computationally efficient while being constrained to be $SE(3)$-equivariant, perfectly suitable for solving complex 3D manipulation tasks under reinforcement learning settings; **iii)** a theoretical justification and extensive empirical analysis for our design choices.

## 2 BACKGROUND

**Message Passing Neural Networks (MPNN)**  Consider a graph $\mathcal{G} = (\mathcal{V}, \mathcal{E})$, where $\mathcal{V}$ represents the nodes and $\mathcal{E}$ the edges. In a standard Graph Neural Network (GNN) (Battaglia et al., 2018), each node $v \in \mathcal{V}$ updates its feature representation by aggregating information from its neighbors $N(v)$. This process is formalized as

$$\mathbf{f}_v^{(k+1)} = \phi\left(\mathbf{f}_v^{(k)}, \bigoplus_{u \in N(v)} \psi\left(\mathbf{f}_v^{(k)}, \mathbf{f}_u^{(k)}, \mathbf{e}_{uv}\right)\right), \tag{1}$$

where $\mathbf{f}_v^{(k)}$ is the feature vector of node $v$ at iteration $k$, $\mathbf{e}_{uv}$ is the edge feature between nodes $u$ and $v$, $\phi$ and $\psi$ are often deep neural networks, and $\bigoplus$ represents an aggregation function like summation, mean or max.

$SE(3)$ **Equivariance and Invariance**    In this work, we focus on geometric graphs $\mathcal{G} = (\mathcal{V}, \mathcal{E}, \mathcal{X})$, where each node $v$ is associated with coordinates $\boldsymbol{x}_v \in \mathcal{X} = \mathbb{R}^3$ and steerable geometric features $\mathbf{f}_v$. A feature is considered *steerable* if it transforms consistently under the action of a group $g \in G$ through a group representation $\rho$. For instance, a vector $\boldsymbol{v} \in \mathbb{R}^3$ under a rotation $\boldsymbol{R} \in SO(3)$ transforms as $\boldsymbol{v}' = \boldsymbol{R}\boldsymbol{v}$. In this case, $\boldsymbol{v}$ is a finite-dimensional vector, and its transformation is described by an invertible matrix representation with determinant 1, $\rho(g) = \boldsymbol{R}$.

Equivariance and invariance are two fundamental concepts in this context, can be formalized as follows, using the notation from Brandstetter et al. (2022):

**Definition 2.1.** *Let $G$ be a group with representations $\rho^{\mathcal{X}}$ and $\rho^{\mathcal{Y}}$. A function $f : \mathcal{X} \to \mathcal{Y}$ is equivariant if*

$$\rho^{\mathcal{Y}}(g)[f(x)] = f(\rho^{\mathcal{X}}(g)[x]), \quad \forall g \in G, x \in \mathcal{X},$$

*and invariant if*

$$f(x) = f(\rho^{\mathcal{X}}(g)[x]), \quad \forall g \in G, x \in \mathcal{X}.$$

In simpler terms, equivariance guarantees that applying a transformation $g$ to the input space $\mathcal{X}$ and then applying the function $f$ produces the same result as applying $f$ first and then transforming the output space $\mathcal{Y}$. On the other hand, invariance implies that the function $f$ remains unchanged when the input undergoes a transformation in $\mathcal{X}$.

**Symmetries in MDPs**    A Markov Decision Process (MDP) is defined by the tuple $(S, A, P, R, \gamma)$, where $S$ is the set of states, $A$ the actions, $P(s'|s, a)$ the transition probability, $R(s, a)$ the reward function, and $\gamma \in [0, 1]$ the discount factor. The goal is to find a policy $\pi : S \to A$ that maximizes the expected discounted reward $\mathbb{E}_\pi[\sum_{t=0}^{\infty} \gamma^t R(s_t, a_t)]$.

In MDPs with symmetries, both the transition distribution $P(s'|s, a)$ and policy distribution $\pi(a|s)$ are invariant under group transformations $g \in G$ via *left-regular representation $L_g$ and $K_g$ for state and action, respectively, resulting in the following conditions:

$$P(L_g[s']|L_g[s], K_g^s[a]) = P(s'|s, a), \quad \pi(K_g^s[a]|L_g[s]) = \pi(a|s),$$

and similarly for the reward function: $R(L_g[s], K_g^s[a]) = R(s, a)$. This allows leveraging symmetries to reduce the complexity of learning, potentially improving sample efficiency and generalization, as it results in a group-structured MDP homomorphism (Van der Pol et al., 2020).

## 3    METHODOLOGY

**Problem Statement**    We aim to solve robotic manipulation problems using an on-policy actor-critic reinforcement learning approach. To address the symmetries present in the state and action spaces of the Markov Decision Process (MDP), we leverage *equivariant policies*, ensuring that transformations applied to the state space are consistently reflected in the action space. To handle the complexities of robotic manipulation, where actuators and objects play distinct roles, we propose the *Heterogeneous Equivariant Policy (HEPi)*, which comprises three key components:

- *Equivariant MPN backbone*: An efficient and expressive EMPN capable of exploiting environment symmetries, thereby significantly reducing the search space complexity.
- *Heterogeneous graph design and update rules*: A graph structure with distinct actuator and object nodes, with tailored message-passing rules to handle the system's heterogeneity.
- Employing a *principled trust-region method* to stabilize training in complex, high-dimensional environments.

### 3.1    EQUIVARIANT MPN BACKBONE

An Equivariant Message Passing Network (EMPN) can be constructed (Brandstetter et al., 2022; Duval et al., 2023; Bekkers et al., 2024) by enforcing equivariance in the functions $\phi$ and $\psi$ in

Equation 1, ensuring that their inputs and outputs are steerable and transform consistently under the group $G$, as described in Definition 2.1. Constructing such functions for high-dimensional steerable features is challenging and typically requires spherical harmonics embeddings (Duval et al., 2023), where matrix-vector multiplications are carried out using Clebsch-Gordan tensor products, followed by steerable activation functions (Brandstetter et al., 2022; Bekkers et al., 2024). While these operations guarantee equivariance, they also introduce high computational complexity, making them impractical for reinforcement learning settings. To mitigate this issue, Bekkers et al. (2024) introduced the PONITA framework, an efficient equivariant message-passing approach. We use it as our EMPN backbone and refer to it as EMPN throughout the rest of the paper for consistency.

Consider the message function in Equation 1, where $\psi(\mathbf{f}_v^{(k)}, \mathbf{f}_u^{(k)}, \mathbf{e}_{uv}) = k(\boldsymbol{x}_u - \boldsymbol{x}_v)\mathbf{f}_u$ is defined as a linear function with steerable feature $\mathbf{f}_u$, and summation is used as the aggregation function, $\bigoplus = \sum$. Here, $k$ is a convolution kernel, and $(\boldsymbol{x}_u - \boldsymbol{x}_v)$ represents the relative position between nodes $u$ and $v$. This leads to the convolutional message-passing update rule, $\mathbf{f}_v' = \sum_{u \in N(v)} k(\boldsymbol{x}_u - \boldsymbol{x}_v)\mathbf{f}_u$. On a regular grid, e.g., an image, each relative position $(\boldsymbol{x}_u - \boldsymbol{x}_v)$ has a corresponding weight $\boldsymbol{W}_{u,v}$ and is stored in one single matrix $\boldsymbol{W}$. However, this does not apply to non-uniform grids, e.g. point clouds, $k(\boldsymbol{x}_u, \boldsymbol{x}_v)$ in this case can be parameterized by a neural network, resulting in the formulation $\mathbf{f}_v' = \int_{\mathcal{X}} k_\theta(\boldsymbol{x}_u, \boldsymbol{x}_v)\mathbf{f}_u d\boldsymbol{x}_u$. Under this general formulation, Bekkers et al. (2024) showed that $k$ can be made $SE(3)$ equivariant by "lifting" the input domain from $\mathcal{X} = \mathbb{R}^3$ to $\mathcal{X}^\uparrow = \mathbb{R}^3 \times \mathcal{S}^2$. Specifically, for every position $\boldsymbol{p} \in \mathbb{R}^3$, an associated orientation $\boldsymbol{o} \in \mathcal{S}^2$ is introduced. This allows features in EMPN to be embedded in both spatial and orientation spaces, leading to the following convolutional form:

$$\mathbf{f}_v' = \int_{\mathbb{R}^3} \int_{\mathcal{S}^2} k_\theta([(\boldsymbol{p}_u, \boldsymbol{o}_u), (\boldsymbol{p}_v, \boldsymbol{o}_v)])\mathbf{f}_u d\boldsymbol{p}_u d\boldsymbol{o}_u.$$

Furthermore, to improve computational efficiency, the kernel function is factorized as:

$$k_\theta([(\boldsymbol{p}_u, \boldsymbol{o}_u), (\boldsymbol{p}_v, \boldsymbol{o}_v)]) = K_\theta^{(3)} k_\theta^{(2)}(\boldsymbol{o}_v^\top \boldsymbol{o}_u) k_\theta^{(1)}(\boldsymbol{o}_v^\top (\boldsymbol{p}_u - \boldsymbol{p}_v), \|\boldsymbol{o}_v \perp (\boldsymbol{p}_u - \boldsymbol{p}_v)\|),$$

where $k^{(1)}$ handles spatial interactions based on the relative position $(\boldsymbol{p}_u - \boldsymbol{p}_v)$ and the perpendicular component $\|\boldsymbol{o}_v \perp (\boldsymbol{p}_u - \boldsymbol{p}_v)\|$, $k^{(2)}$ manages orientation-based interactions via dot products $\boldsymbol{o}_v^\top \boldsymbol{o}_u$, and $K^{(3)}$ performs channel-wise mixing across features. This formulation preserves the universal approximation property of equivariant functions while being significantly more computationally efficient and not requiring specialized network structures. Furthermore, $\boldsymbol{o}$ can be sampled on a uniform grid over $\mathcal{S}^2$, making EMPN only approximately equivariant. However, in practice, it achieves strong performance even with a limited number of samples, as discussed in Appendix D.

## 3.2 Heterogeneous Equivariant Policy

In robotic manipulation tasks, actuators and objects play fundamentally distinct roles. The graph is defined as $\mathcal{G} = (\mathcal{V}, \mathcal{E})$, where $\mathcal{V} = \mathcal{V}_{\text{act}} \cup \mathcal{V}_{\text{obj}}$ represents disjoint node sets for actuators and objects. Our approach captures these roles by first processing local information within the object and actuator clusters and then aggregating it globally to the actuators via directed, fully-connected inter-edges, as shown in Figure 1. This design distinguishes object-to-object, actuator-to-actuator, and object-to-actuator interactions, allowing the system to separate local processing from global information exchange.

The updates for both object and actuator nodes can be expressed as

$$\mathbf{f}_v^{\text{obj, new}} = \phi_{\text{obj}}\left(\mathbf{f}_v^{\text{obj}}, \sum_{u \in N(v)_{\text{obj}}} k(x_u^{\text{obj}}, x_v^{\text{obj}}; \theta_{\text{obj-obj}})\mathbf{f}_u^{\text{obj}}\right), \quad v \in \mathcal{V}_{\text{obj}},$$

$$\mathbf{f}_v^{\text{act, new}} = \phi_{\text{act-local}}\left(\mathbf{f}_v^{\text{act}}, \sum_{w \in N(v)_{\text{act}}} k(x_w^{\text{act}}, x_v^{\text{act}}; \theta_{\text{act-act}})\mathbf{f}_w^{\text{act}}\right), \quad v \in \mathcal{V}_{\text{act}}, \tag{2}$$

$$\mathbf{f}_v^{\text{act, final}} = \mathbf{f}_v^{\text{act, new}} + \phi_{\text{act-global}}\left(\mathbf{f}_v^{\text{act}}, \sum_{u \in \mathcal{V}_{\text{obj}}} k(x_u^{\text{obj}}, x_v^{\text{act}}; \theta_{\text{obj-act}})\mathbf{f}_u^{\text{obj, new}}\right), \quad v \in \mathcal{V}_{\text{act}}.$$

Here, $\mathbf{f}_v^{\text{obj, new}}$ represents the updated object features after local object-to-object interactions, $\mathbf{f}_v^{\text{act, new}}$ refers to the updated actuator features after local actuator-to-actuator interactions, and $\mathbf{f}_v^{\text{act, final}}$ is the final feature for actuator nodes after aggregating information from both the objects and its actuator neighbors. Here each of the kernels $k(\cdot, \cdot; \theta_{\text{obj-obj}})$, $k(\cdot, \cdot; \theta_{\text{obj-act}})$, and $k(\cdot, \cdot; \theta_{\text{act-act}})$, has it own learnable parameters, allowing them to specialize the learning process for each interaction type.

Moreover, each node $v \in \mathcal{V}$ encodes its `node_type` as a one-hot scalar-vector, along with normalized position vectors $\mathbf{p}_v$ and velocities $\mathbf{v}_v$. For object nodes, the feature vector also includes the relative distance to the target, $\mathbf{d}_{v,\text{target}}$, embedding target information directly without the need for an additional target node. For actuator nodes, the output consists of both a scalar $c$ and a vector $\mathbf{v}_{\text{out}}$, where the final output vector is computed as $\mathbf{v}_{\text{out}} = c \cdot \mathbf{v}$. This setup ensures flexibility for diverse tasks while maintaining consistency with the geometric properties of the system.

**Value Function** We employ DeepSets (Zaheer et al., 2017) with the same input structure as the policy to preserve permutation invariance of the node features, while keeping the architecture both simple and computationally efficient, similar to the prior work Simm et al. (2021). The value function is computed as $V(s) = \text{MLP}_{\text{outer}}\left(\sum_{v \in \mathcal{V}} \text{MLP}_{\text{inner}}(s_v)\right)$, where $s_v$ represents the feature of node $v$. These node features may differ from those used in the policy network to capture task-specific observations. For example, in the *Cloth-Hanging* task, the value function considers features from all nodes, while the policy network focuses only on the hole boundary nodes. Full details of the input features used for the value function are provided in the Appendix B.

**Trust-Region Projection Layers** Standard on-policy reinforcement learning approaches such as Proximal Policy Optimization (PPO) (Schulman et al., 2017), learn a policy by optimizing the surrogate objective

$$\theta_{k+1} = \arg\max_\theta \mathbb{E}_{(s,a) \sim \pi_{\theta_k}}\left[\frac{\pi_\theta(a|s)}{\pi_{\theta_k}(a|s)} A^{\pi_{\theta_k}}(s,a)\right] \quad \text{s.t.} \quad D_{\text{KL}}(\pi_\theta || \pi_{\theta_k}) \leq \delta,$$

where $A^{\pi_{\theta_k}}(s,a)$ is the advantage function. Here, $D_{\text{KL}}(\pi_\theta || \pi_{\theta_k})$ is the KL-divergence between the new policy $\pi_\theta$ and the old policy $\pi_{\theta_k}$, constrained by $\delta$ to ensure stable updates and prevent overly large policy changes. PPO approximates this trust region by clipping the importance sampling ratio to limit updates. This, however, requires careful hyperparameters turning to make it work stably, as pointed out by Andrychowicz et al. (2021). We will show later in the Results Section 4.2, this is also applied to graph-based policy. On the other hand, Trust Region Projection Layers (TRPL) (Otto et al., 2021) adopt a more principled approach. TRPL projects policy parameters onto trust region boundaries using a differentiable convex optimization, ensuring stability by projecting both the mean and variance of the Gaussian policy to satisfy trust region constraints.

In this paper, we adopt TRPL to ensure stable policy updates, and we will show in the Result Section 4.2, TRPL consistently outperforms PPO, with little hyperparameter tuning required, especially in tasks requiring complex exploration, as also observed in the prior works (Otto et al., 2023; Li et al., 2023; Celik et al., 2024).

### 3.3 THEORETICAL JUSTIFICATION

HEPi is inspired by adding global Virtual Nodes ($\text{VN}_G$) to Message Passing Neural Networks, (MPNNs). For example, Southern et al. (2024) proposed MPNN + $\text{VN}_G$ which seperates local and global updates. Here, the local updates are equivalent to our object node updates, and the global ones correspond to our actuator node update. Based on this interpretation, we show that locally connecting actuator nodes to only k-nearest object nodes can not capture relevant relation between object and actuator nodes, while treating the actuator nodes as VN that connects to all object nodes can. We name the graph network with locally connected actuators and object nodes as MPNN + $\text{VN}_{\text{Local}}$. We show that for HEPi any two actuator and object nodes can exchange information, while this is not the case for MPNN + $\text{VN}_{\text{Local}}$.

**Proposition 3.1.** *For MPNN + $VN_{Local}$, the Jacobian $\partial \mathbf{f}_v^{act} / \partial \mathbf{f}_u^{obj}$ is independent of $u$ whenever object node $u$ and actuator node $v$ are separated by more than 2 hops. In contrast, HEPi with node connections and updates as described in Section 3.2 can exchange information between any actuator and object nodes after a single layer.*

The proof is provided in Appendix A. This result implies that HEPi's connection design allows the actuators to receive relevant information to predict actions w.r.t changes at object nodes. In contrast,

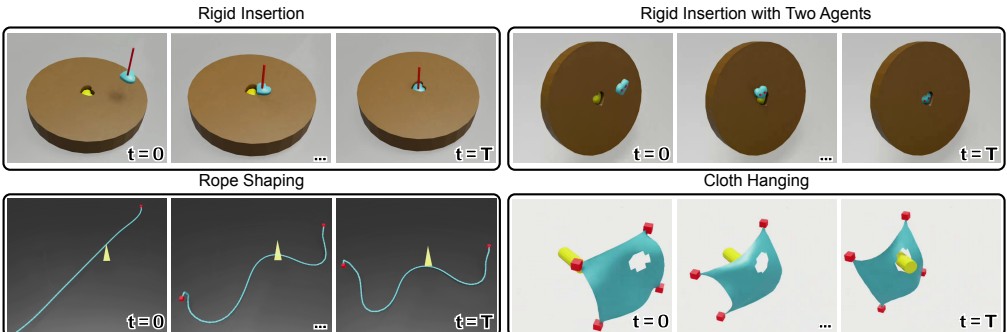

Figure 2: Illustration of our diverse and challenging manipulation tasks, involving both rigid and deformable objects. These tasks require precise control under complex geometric constraints, co-ordination between multiple actuators, and handling of intricate interactions between objects and actuators. The variety of tasks highlights the need for policies that can understand the geometric structure in large observation and action spaces.

for MPNN + VN$_{\text{Local}}$ the actuators could fail to predict relevant actions to changes at object node $u$. We provide experimental ablations across various values of $k$ to clearly emphasize this distinction in Appendix D.

## 4    EXPERIMENTS

In this section, we outline the experimental setup and present the results comparing the proposed HEPi against other baselines.

### 4.1    EXPERIMENTAL SETUP

**Task Design**    Our task design, illustrated in Figure 2, emphasizes testing the role of geometric structure and information exchange between objects and actuators in robotic manipulation. To focus on this, we abstract away the specifics of the robot body and consider only end-effector control. We introduce two categories of tasks: rigid manipulation on diverse geometries and deformable object manipulation, all implemented in NVIDIA IsaacLab (Mittal et al., 2023) to leverage its GPU-based parallelization capabilities[1].

The rigid manipulation tasks are inspired by Transporter Net (Zeng et al., 2020). **Rigid-Sliding** mimics using a suction gripper to slide an object across a 2D plane to a target position and orientation, with 10 distinct objects, and randomized initial and target poses. Next, **Rigid-Pushing** removes the physical connection between the actuator and the object, allowing the actuator to move freely in the $x$-$y$ plane to push the object to a desired target position and orientation. **Rigid-Insertion**, similar to the assembly kit task in Transporter Net, extends this to 3D, requiring precise alignment and insertion of objects into holes, using 8 different objects. Additionally, we introduce a novel **Rigid-Insertion-Two-Agents** task, where two linear actuators work together to control an object, guiding it to a target randomly positioned in the upper hemisphere of the $\mathcal{S}^2$.

For deformable object manipulation, we first adopt the **Rope-Closing** task from Laezza et al. (2021), where two actuators manipulate a deformable rope to wrap around a cylindrical object in a 2D plane, with randomized initial configurations. We then introduce a novel task, **Rope-Shaping**, which increases complexity by requiring the rope to form a specific shape (a "W" from the LASA dataset (Khansari-Zadeh & Billard, 2011)) to a desired orientation. Finally, we introduce **Cloth-Hanging**, where four actuators control the corners of a cloth to hang it onto a hanger, with randomized starting positions and orientations in 3D space.

These tasks present a range of manipulation challenges, emphasizing the role of geometric structure and requiring complex exploration strategies to coordinate the agents in completing the tasks. Full task details, including reward definitions, are provided in Appendix B.

---

[1]A video showcasing the tasks can be found in the supplementary material.

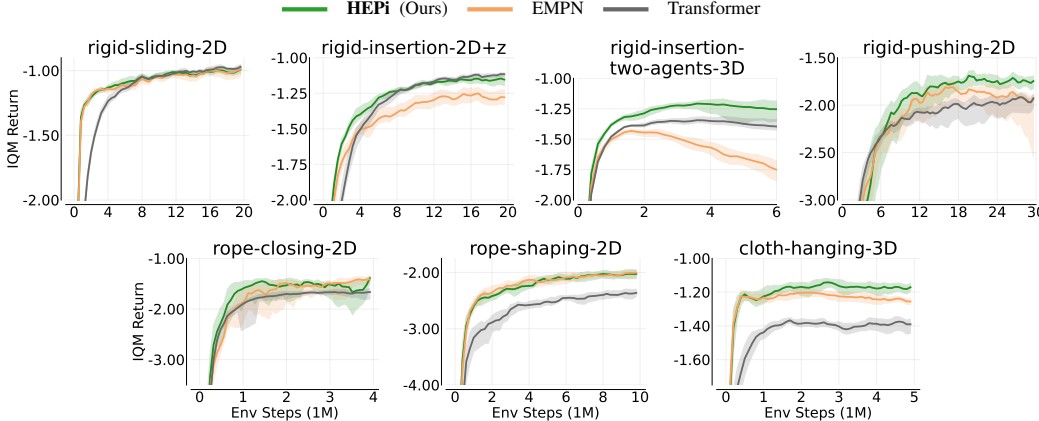

Figure 3: Evaluation curves for our seven manipulation tasks, comparing HEPi (ours), EMPN, and Transformer baselines. Results are averaged over 10 seeds, using IQM with 95% confidence intervals. HEPi consistently outperforms EMPN and Transformer in tasks requiring complex exploration and heterogeneity handling, such as *rigid-insertion-two-agents-3D*, *rigid-pushing-2D* and *cloth-hanging-3D*.

**Baselines**   We compare HEPi against two primary baselines: policies based on a Transformer (Vaswani et al., 2017) and a naive EMPN. Transformers serve as a strong baseline to evaluate in our setting as it can be seen as a fully-connected GNN (Battaglia et al., 2018), and have achieved state-of-the-art performance in other graph-based reinforcement learning problems (Kurin et al., 2021; Trabucco et al., 2022; Gupta et al., 2022; Hong et al., 2022). In addition, for the *Cloth-Hanging* task, we evaluate two additional baselines, Heterogeneous GNN (HeteroGNN) and a naive GNN to highlight the effectiveness of incorporating equivariant constraints in a large 3D space.

Our experimental setup aims to answer the following key questions: **(1)** Can explicitly modeling heterogeneity between actuators and objects, combined with $SE(3)$ equivariance, improve performance in complex 3D tasks? **(2)** How well does HEPi generalize when dealing with different geometries, resolutions in rigid tasks, and varying sample spaces in the complex 3D *cloth-hanging* task? **(3)** Attention mechanisms are often employed in GNNs to capture heterogeneity, do they offer the same benefits as explicitly modeling heterogeneity in HEPi? **(4)** Does using trust-region methods in HEPi stabilize the training process more effectively than a naive PPO?

## 4.2   RESULTS AND DISCUSSIONS

In the main evaluations, we generate 1000 scenes per task (sampled according to Appendix B) and compute the undiscounted return over 10 seeds, and report the average using Interquartile Mean (IQM) (Agarwal et al., 2021) with 95% confident interval. Figure 3 shows the evaluation curves for the seven manipulation tasks. Overall, both EMPN and HEPi outperform the Transformer in terms of sample complexity, owing to their ability to exploit symmetry.

For final performance on rigid tasks, firstly in *rigid-slding-2D* and *rigid-insertion-2D+z* tasks, HEPi and Transformer policies perform comparably, suggesting that the limited task complexity does not fully leverage the benefits of equivariant constraints. However, when the search space grows larger, as in the case of *rigid-insertion-two-agents-3D*, Transformer struggles to find a policy that generalizes to all poses. EMPN, on the other hand, gets stuck in local optima due to its lack of expressiveness, especially in tasks requiring more exploration, such as *rigid-pushing-2D*, *rigid-insertion-2D+z* and *rigid-insertion-two-agents-3D*. In contrast, HEPi's explicit handling of heterogeneity allows for more effective exploration, leading to better overall performance.

In *rope-closing* and *rope-shaping*, simple deformable object tasks, the Transformer exhibits poor generalization, likely due to the complexity introduced by non-rigid constraints and random orientations. HEPi and EMPN perform similarly on the 2D tasks, but as tasks scale up to 3D environments, such as *cloth-hanging-3D*, HEPi shows a significant advantage, outperforming both baselines. This highlights the importance of explicitly capturing heterogeneity and task geometry in manipulation.

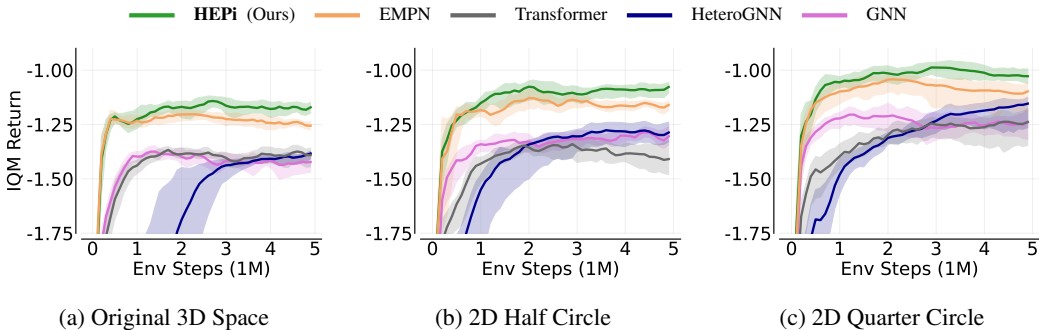

(a) Original 3D Space      (b) 2D Half Circle      (c) 2D Quarter Circle

Figure 4: Performance of different models on the *Cloth-Hanging* task across varying sample spaces. Overall, performance improves as the sample space decreases. In terms of final performance, heterogeneous models outperform homogeneous baselines in most cases, demonstrating the benefits of explicit heterogeneity modeling. Additionally, applying equivariant constraints is critical for achieving superior performance in 3D tasks. More results can be found in Appendix D.

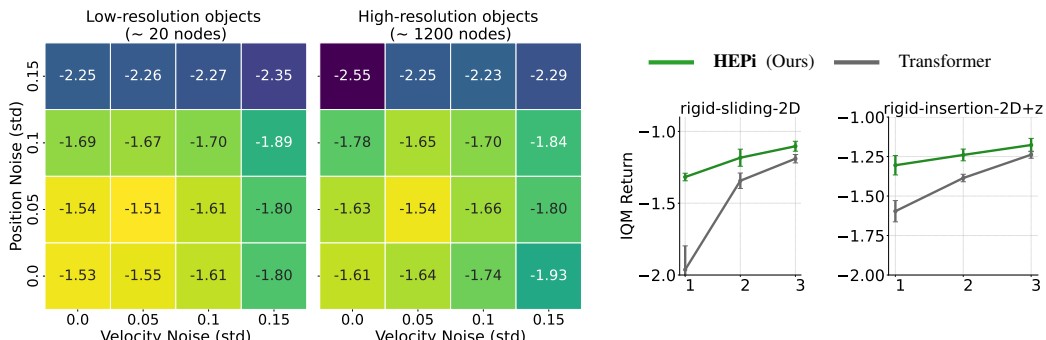

Figure 5: **Left:** Analysis of noise sensitivity and scalability to high-resolution objects in the *Rigid-Pushing* task. Heatmaps show average returns under varying levels of artificial Gaussian noise in position and velocity inputs for both low-resolution and high-resolution objects. A single HEPi agent, trained on a low-resolution with additive Gaussian Noise ($\sigma = 0.01$), was used for all evaluations. **Right:** Generalization performance on *Rigid-Sliding* and *Rigid-Insertion* tasks. Models are trained on one object (*plus*), two objects (*plus*, *star*), and three objects (*plus*, *star*, *pentagon*) and tested on the remaining unseen objects. Overall, HEPi generalizes well to unseen objects, performs consistently across resolutions, and handles noise effectively, making it suitable for real-world tasks.

**Generalizability** Figure 4 compares the performance of different models on the *Cloth-Hanging* task across varying sample spaces. As expected, smaller sample spaces simplify the exploration and improve performance. Heterogeneous models (HeteroGNN and HEPi) consistently achieve higher final returns than their homogeneous counterparts, demonstrating greater expressiveness. However, HeteroGNN requires more samples, whereas HEPi's use of EMPN significantly improves sample efficiency by leveraging equivariant constraints in large 3D spaces.

Next, we evaluate the robustness of HEPi to noisy inputs and its ability to handle high-resolution object meshes on the *Rigid-Pushing* task. GNNs naturally capture locality through message passing, allowing them to scale effectively to higher-resolution graphs without retraining (Li et al., 2020; Freymuth et al., 2023). During training, we added Gaussian noise ($\sigma = 0.01$) to normalized positions and velocities to encourage diverse node representations, a common GNN regularization technique (Godwin et al., 2022). The best HEPi agent was then evaluated with varying Gaussian noise levels applied to pre-normalized inputs (environment noise) and across low-resolution ($\sim 20$ nodes) and high-resolution ($\sim 1200$ nodes) object meshes. As shown in Figure 5 (left), HEPi maintains high performance across resolutions with only mild degradation at higher noise levels, demonstrating its scalability and robustness to noisy and diverse object representations.

Finally, we evaluate the generalization of these models to unseen objects on two rigid tasks: *rigid-sliding* and *rigid-insertion*. Both tasks are trained on subsets of objects—one (*plus*), two (*plus*, *star*),

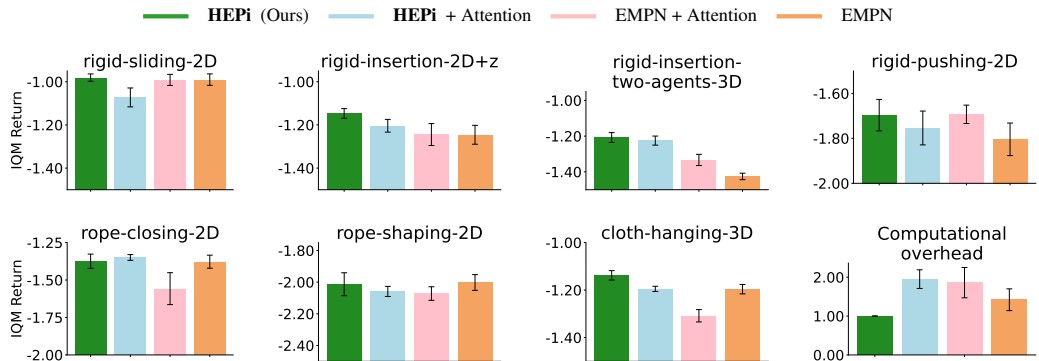

Figure 7: Performance comparison on tasks with and without attention mechanisms over 10 seeds. Adding attention significantly increases the training time but does not improve performance, as shown on the right. The computational overhead is measured as the ratio of training time per iteration (over seven tasks) relative to HEPi.

and three (*plus*, *star*, *pentagon*)—and tested on the remaining objects. Figure 5 (right) shows that HEPi generalizes better than the Transformer baseline, benefiting from the ability of graph-based models to exploit object topology. In contrast, Transformers lack structure-aware embeddings and struggle with graph-structured inputs, as noted in prior work (Hong et al., 2022).

**Attention** Attention mechanisms are widely used in graph neural networks to capture heterogeneity. In this experiment, we examine the impact of adding attention as an aggregation function in Equation 1 to both homogeneous and heterogeneous graph networks, framing the popular Graph Attention Network (GAT) framework (Veličković et al., 2018). However, as shown in Figure 7, attention does not provide any noticeable benefit across the tasks.

While attention helps capture some heterogeneity, particularly in tasks like *rigid-pushing-2D* and *rigid-insertion-two-agents-3D*, it ultimately complicates the learning process in on-policy reinforcement learning, making the optimization landscape more difficult to traverse. Additionally, adding attention significantly increases training time without improving performance, e.g., for HEPi it almost doubled.

**Training Stability** To investigate the impact of the TRPL method, we compare it against PPO. For a fair comparison, we perform a grid search over the `clip_eps` parameter in PPO (Appendix E.3). Overall, as depicted in Figure 8, in tasks requiring high exploration such as *cloth-hanging-3D*, PPO struggles to maintain conservative updates, often resulting in unstable performance. However, in tasks with a lower-dimensional action space, such as 2D environments, well-tuned PPO performs comparably to TRPL in terms of final average return, though being less sample efficient. This suggests that while PPO can be tuned to perform adequately in simpler action spaces, TRPL provides more stability and robustness, particularly in complex 3D environments that demand more effective exploration control.

## 5 RELATED WORK

**Equivariant Policies for Robotic Manipulation** In imitation learning for robotic manipulation, equivariance has been widely applied to reduce the effort of collecting human demonstrations (Zeng et al., 2020; Huang et al., 2022; 2024; Ryu et al., 2023; Yang et al., 2024). Most prior work focuses on simple pick-and-place tasks, where the policy outputs the 3D pose of the end-effector. By leveraging equivariance, these policies can generalize across different object poses, significantly reducing the number of required demonstrations (e.g., only 5 to 10 demonstrations in (Ryu et al., 2023)). However, 3D pose actions are insufficient for tasks that require higher dexterity or involve deformable objects. To address this limitation, Equibot (Yang et al., 2024) designed an equivariant policy that outputs velocity vectors, achieving success in more complex tasks such as cloth folding and object wrapping.

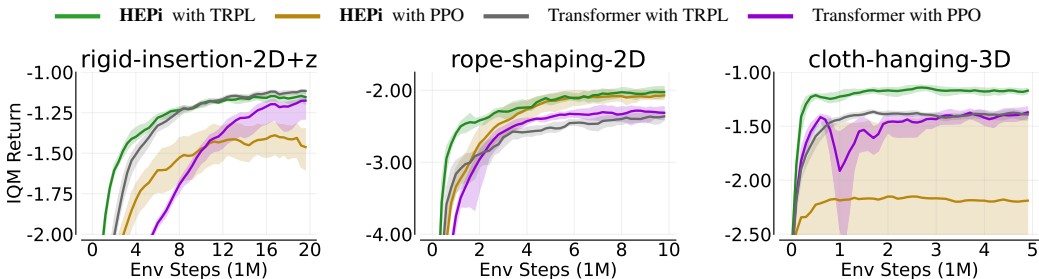

Figure 8: Performance comparison between HEPi and Transformer models with TRPL and PPO over 10 seeds. TRPL shows stable performance across all tasks, while PPO struggles in tasks requiring high exploration, especially in 3D environments like *cloth-hanging-3D*. In tasks with a lower-dimensional action space (e.g., 2D tasks), both methods perform comparably when carefully tuned.

There has been limited work on exploiting equivariant policies in reinforcement learning, and existing approaches have largely focused on 2D spaces (Wang et al., 2022; Nguyen et al., 2023). In this work, we extend the study of equivariant policies to 3D space within a reinforcement learning setting, which, to the best of our knowledge, has not yet been explored for robotic manipulation.

**RL with GNNs**  Graph-based representations in reinforcement learning have shown great success across diverse domains, including molecular design (Simm et al., 2021), adaptive mesh refinement (Freymuth et al., 2023), and multi-agent systems like traffic light control (van der Pol et al., 2022). Among these, the most closely related work to ours is morphology reinforcement learning (Wang et al., 2018; Huang et al., 2020; Trabucco et al., 2022; Hong et al., 2022; Gupta et al., 2022), where the robot's kinematic structure is represented as a graph, allowing actuators to be controlled through message passing. This approach enables policies to generalize across different robot topologies, particularly in locomotion tasks (Gupta et al., 2022). Chen et al. (2023) extend these ideas to handle 3D environments, but only with sub-equivariant policies for rotations around the gravity axis.

However, these works primarily exploit the locality of graph structures, framing the problem as multi-agent reinforcement learning on graphs (Jiang et al., 2020), where each node can make decisions influenced by its neighbors. In contrast, our work focuses on the underactuated problem where only a small subset of nodes (actuators) controls a much larger set of object nodes, framing a heterogeneous graph. The interactions between the nodes are therefore much more complex to capture using homogeneous GNN models.

## 6  CONCLUSION

We have demonstrated that robotic manipulation problems can be effectively represented as heterogeneous graphs, comprising two sub-graphs to capture the geometric structure of the environment. Building on this, we introduced HEPi, a graph-based policy featuring multiple equivariant message-passing networks as its backbone. These networks are constrained to be equivariant under $SE(3)$ transformations, which significantly improves sample efficiency. Furthermore, HEPi explicitly models heterogeneity by assigning distinct network parameters for each interaction type, reducing message mixing and improving expressiveness. This approach has proven less prone to converging on sub-optimal solutions. To assess the effectiveness of our approach, we developed a new reinforcement learning benchmark focused on manipulating objects with diverse geometries and deformable materials. Our results show that HEPi outperforms both the state-of-the-art Transformer and its non-heterogeneous, non-equivariant counterparts.

**Limitation**  In our current setup, we abstract away the robot body, focusing solely on end-effector movements. Future work could explore incorporating a more structured representation of actuator nodes, potentially leveraging the robot's full morphology. Moreover, although our approach does not require full object meshes, we assume that the keypoint coordinates are readily available as our main observation. This limitation could be addressed by integrating state-of-the-art computer vision techniques to extract keypoints from cameras (Tumanyan et al., 2024; Hou et al., 2024), using these as object nodes, thus increasing its applicability in real-world scenarios.

ACKNOWLEDGMENTS

This work is part of the M4 subproject of the DFG AI Research Unit 5339, which focuses on designing distributed policy optimization methodologies to accelerate manufacturing process maturation. We gratefully acknowledge financial support from the German Research Foundation (DFG) and computational resources provided by bwHPC and the HoreKa supercomputer, funded by the Ministry of Science, Research, and the Arts Baden-Württemberg and the German Federal Ministry of Education and Research.

We thank Niklas Freymuth, Philipp Dahlinger, Onur Celik, Aleksandar Taranovic, and Mayank Mittal for their insightful discussions during both the initial and rebuttal phases of this paper, as well as for their technical support. We also appreciate Reviewer 6nhr for suggesting the Rigid-Pushing task.

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

# A    PROOFS OF PROPOSITION 3.1

*Proof.* We write simplified updates of HEPi and MPNN + VN$_{\text{local}}$ as follows,

**HEPi**:

$$\mathbf{f}_v^{\text{obj},(l+1)} = \mathbf{f}_v^{\text{obj},(l)} + \sigma \left( W_o^{(l)} \sum_{u \in N(v)_{\text{obj}}} k(\cdot, \cdot; \theta_{\text{obj-obj}}) \mathbf{f}_u^{\text{obj},(l)} \right), \quad v \in \mathcal{V}_{\text{obj}},$$

$$\mathbf{f}_v^{\text{act, new},(l+1)} = \mathbf{f}_v^{\text{act},(l)} + \sigma \left( W_a^{\text{local},(l)} \sum_{w \in N(v)_{\text{act}}} k(\cdot, \cdot; \theta_{\text{act-act}}) \mathbf{f}_w^{\text{act},(l)} \right), \quad v \in \mathcal{V}_{\text{act}}, \tag{3}$$

$$\mathbf{f}_v^{\text{act},(l+1)} = \mathbf{f}_v^{\text{act, new},(l+1)} + \mathbf{f}_v^{\text{act},(l)} + \sigma \left( W_a^{(l)} \sum_{u \in \mathcal{V}_{\text{obj}}} k(\cdot, \cdot; \theta_{\text{obj-act}}) \mathbf{f}_u^{\text{obj, L}} \right), \quad v \in \mathcal{V}_{\text{act}}.$$

where $\sigma$ is activation function, and $W^{(l)}$ is the weight at layer $l$. Note that the object nodes are updated through $L$ layers.

**MPNN + VN$_{\text{local}}$**:

$$\mathbf{f}_v^{\text{obj},(l+1)} = \mathbf{f}_v^{\text{obj},(l)} + \sigma \left( W_o^{(l)} \sum_{u \in N(v)_{\text{obj}}} k(\cdot, \cdot; \theta_{\text{obj-obj}}) \mathbf{f}_u^{\text{obj},(l)} \right), \quad v \in \mathcal{V}_{\text{obj}},$$

$$\mathbf{f}_v^{\text{act, new},(l+1)} = \mathbf{f}_v^{\text{act},(l)} + \sigma \left( W_a^{\text{local},(l)} \sum_{w \in N(v)_{\text{act}}} k(\cdot, \cdot; \theta_{\text{act-act}}) \mathbf{f}_w^{\text{act},(l)} \right), \quad v \in \mathcal{V}_{\text{act}}, \tag{4}$$

$$\mathbf{f}_v^{\text{act},(l+1)} = \mathbf{f}_v^{\text{act, new},(l+1)} + \mathbf{f}_v^{\text{act},(l)} + \sigma \left( W_a^{(l)} \sum_{u \in N_k(v)_{\text{obj}}} k(\cdot, \cdot; \theta_{\text{obj-act}}) \mathbf{f}_u^{\text{obj}, L} \right), \quad v \in \mathcal{V}_{\text{act}}.$$

As seen, the main difference between HEPi and MPNN + VN$_{\text{local}}$ is at treating the actuator nodes as VN nodes as in MPNN + VN$_G$ or normal graph nodes with k-NN connections.

For HEPi, the actuator nodes are updated through every object node as in the third equation in Eq. 3. Explicitly, we compute its Jacobian w.r.t object nodes as

$$\frac{\partial \mathbf{f}_v^{\text{act},(l+1)}}{\partial \mathbf{f}_u^{\text{obj}, L}} = 2 \nabla \mathbf{f}_v^{\text{act},(l)} + \sigma'(z_v^{\text{local},(l)}) W_a^{\text{local},(l)} \sum_{w \in N(v)_{\text{act}}} k(x_v, x_w; \theta_{\text{act-act}}) \nabla \mathbf{f}_v^{\text{act},(l)} \tag{5}$$

$$+ \sigma'(z_v^{(l)}) W_a^{(l)} k(x_v, x_u; \theta_{\text{obj-act}})$$

with $z_v^{(l)} = W_a^{(l)} \sum_{u \in \mathcal{V}_{\text{obj}}} k(\cdot, \cdot; \theta_{\text{obj-act}}) \mathbf{f}_u^{\text{obj}, L}$ and $z_v^{\text{local},(l)} = W_a^{\text{local},(l)} \sum_{w \in N(v)_{\text{act}}} k(\cdot, \cdot; \theta_{\text{act-act}}) \mathbf{f}_w^{\text{act},(l)}$ be the evaluation of the argument of the function $\sigma$. This shows that any object node can exchange information with the actuator nodes after a single layer of the object-actuator update.

For MPNN + VN$_{\text{local}}$, if an actuator node $v$ and an object node $u$ are more than 2-hops distant from each other, the message from node $u$ sent to $v$ will arrive either through another actuator node (via actuator-actuator updates) or through a node where the $k$-NN connections of those actuator nodes overlap (depicted in Figure 9). However, in both cases, the Jacobian at the actuator node $v$ becomes independent of the feature at the object node $u$, i.e., it can only receive a homogeneous value from the VN (overlapping node or other actuator node) (Southern et al., 2024). Consequently, the policy could fail to predict relevant actions in response to changes at the object node $u$.    □

**Related Work for Oversquashing in Graph Neural Networks**    Transformers (Vaswani et al., 2017) can be viewed as fully connected GNNs under the Message Passing Neural Network (MPNN) framework (Battaglia et al., 2018), since self-attention can be seen as a mechanism to aggregate

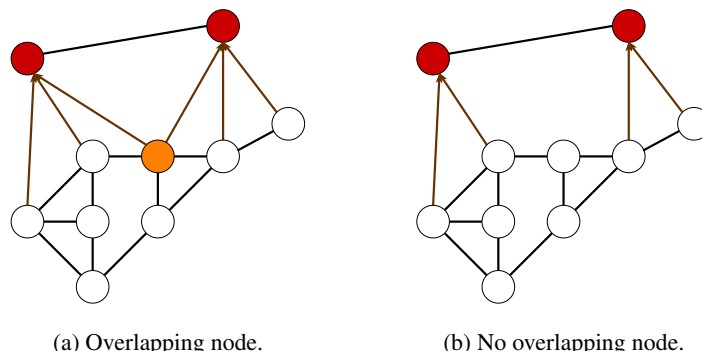

(a) Overlapping node.         (b) No overlapping node.

Figure 9: Demonstration of graph with overlapping and non-overlapping nodes. Actuator nodes are in red, object nodes are in either white or orange.

messages from its neighbors. While GNNs offer efficient local information processing with linear complexity $\mathcal{O}(|V| + |E|)$, they struggle with *over-squashing*, limiting their ability to propagate information across distant nodes (Di Giovanni et al., 2023). In contrast, Transformers, by being fully connected, allow every node to exchange information with all others, making them well-suited for tasks requiring global information aggregation, though at a higher quadratic complexity $\mathcal{O}(|V|^2)$.

Recent studies have shown that introducing virtual nodes (VN) in GNNs can mitigate the over-squashing issue by facilitating long-range information exchange while retaining the lower complexity of GNNs (Di Giovanni et al., 2023; Cai et al., 2023; Rosenbluth et al., 2024). Our HEPi builds on this insight, treating actuators as virtual nodes to enable efficient global information aggregation from object nodes, as shown in Figure 1.

## B  TASKS DETAILS

Here, we provide detailed specifications for each of the seven manipulation tasks introduced in the main paper.

### B.1  RIGID-SLIDING

The goal of the Rigid-Sliding task is to control an object using a suction gripper and slide it on a 2D plane to a desired target position and orientation. The agent controls the object's linear velocity $v$ and angular velocity $\omega$ in the yaw direction.

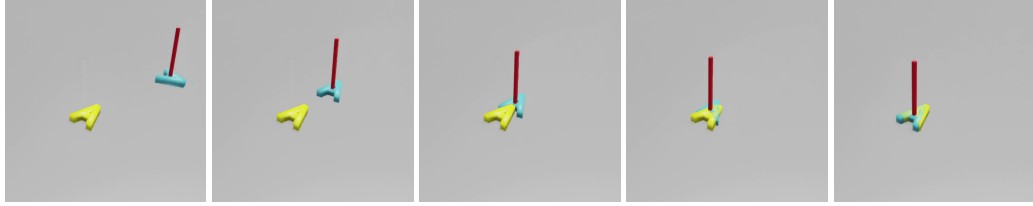

Figure 10: Example trajectory of Rigid Sliding task.

**Input and Output**  The input space for each node includes:

- Gripper nodes: `node_type`, position $\mathbf{p}_a$, velocity $\mathbf{v}_a$, angular velocity $\omega_a$.
- Object nodes: `node_type`, position $\mathbf{p}_o$, distance to target $d_{\text{target}}$.

The output consists of the gripper's linear velocity $v_a$ and a vector from which the angular velocity $\omega_a$ is derived. Specifically, the vector is decomposed into its parallel and tangential components

with respect to the position vector $\mathbf{r}$, where $\mathbf{v}_{\parallel} = \left( \frac{\mathbf{v} \cdot \mathbf{r}}{\|\mathbf{r}\|^2} \right) \mathbf{r}$ and $\mathbf{v}_{\perp} = \mathbf{v} - \mathbf{v}_{\parallel}$. The angular velocity is then computed as $\omega_a = \frac{\mathbf{r} \times \mathbf{v}_{\perp}}{\|\mathbf{r}\|^2}$.

**Sample Space**

- Initial pose: $(x, y, \theta_{\text{yaw}}) \in [-1, 1]^2 \times [-\pi, \pi]$.
- Target pose: $\theta_{\text{yaw}} \in [-\pi, \pi]$.

**Reward Function**    The reward consists of multiple sub-rewards:

- **Distance to goal**:
$$R_{\text{goal}} = \|\mathbf{p}_o - \mathbf{p}_{goal}\|$$
  where $\mathbf{p}_o$ is the object position and $\mathbf{p}_{goal}$ is the target position.
- **Rotation distance**:
$$R_{\text{rotation}} = \texttt{quat\_diff}(\mathbf{r}_o, \mathbf{r}_{goal})$$
  where $\mathbf{r}_o$ and $\mathbf{r}_{goal}$ are the object and goal orientations in quaternion.
- **Object velocity penalty**:
$$V_{\text{object}} = v_{\text{angular}} + v_{\text{linear}}$$
  where $v_{\text{angular}}$ and $v_{\text{linear}}$ are the angular and linear velocities of the object.
- **Action rate penalty**:
$$A_{\text{actions}} = \sqrt{a_i - a_{i-1}}$$
  where $a_i$ and $a_{i-1}$ are the actions at the current and previous time steps.

The total time-dependent reward with $T = 100$ is:

$$R_{\text{tot}} = \begin{cases} -0.8 R_{\text{goal}} - 0.4 R_{\text{rotation}} - 0.1 V_{\text{object}} - 0.002 A_{\text{actions}}, & t < T - 2, \\ -4.0 R_{\text{goal}} - 2.0 R_{\text{rotation}} - 0.1 V_{\text{object}} - 0.002 A_{\text{actions}}, & t \geq T - 2. \end{cases}$$

### B.2    RIGID-PUSHING

The goal of the Rigid-Pushing task is to control an object using a rod and push it on a 2D plane to a desired target position and orientation. The agent controls the object's linear velocity $v$.

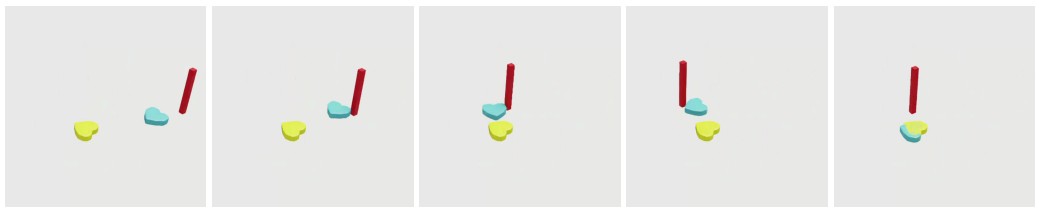

Figure 11: Example trajectory of Rigid Pushing task.

**Input and Output**    The input space for each node includes:

- Gripper nodes: $\texttt{node\_type}$, position $\mathbf{p}_a$, velocity $\mathbf{v}_a$, angular velocity $\omega_a$.
- Object nodes: $\texttt{node\_type}$, position $\mathbf{p}_o$, distance to target $d_{\text{target}}$, and object's linear velocity $\mathbf{v}_o$.

The output consists of the actuator's linear velocity $v_a$.

**Sample Space**

- Initial pose: $(x, y, \theta_{\text{yaw}}) \in [-0.5, 0.5]^2 \times [-\pi, \pi]$.
- Target pose: $\theta_{\text{yaw}} \in [-\pi, \pi]$.

**Reward Function**    The reward consists of multiple sub-rewards:

- **Distance to goal**:
$$R_{\text{goal}} = \|\mathbf{p}_o - \mathbf{p}_{goal}\|$$
  where $\mathbf{p}_o$ is the object position and $\mathbf{p}_{goal}$ is the target position.

- **Rotation distance**:
$$R_{\text{rotation}} = \texttt{quat\_diff}(\mathbf{r}_o, \mathbf{r}_{goal})$$
  where $\mathbf{r}_o$ and $\mathbf{r}_{goal}$ are the object and goal orientations in quaternion.

- **Distance to object**:
$$R_{\text{object}} = \|\mathbf{p}_o - \mathbf{p}_{actuator}\|,$$
  encouraging the rod (actuator) to stay close to the object during pushing.

The total time-dependent reward with $T = 100$ is:

$$R_{\text{tot}} = \begin{cases} -0.8R_{\text{goal}} - 0.08R_{\text{rotation}} - 0.2R_{\text{object}}, & t < T - 5, \\ -8.0R_{\text{goal}} - 0.8R_{\text{rotation}} - 0.2R_{\text{object}}, & t \geq T - 5. \end{cases}$$

### B.3    RIGID-INSERTION

The Rigid-Insertion task extends Rigid-Sliding to 3D, where the agent must control both linear and angular velocities to move an object along the $z$-axis and insert it into a hole. The state space includes $(x, y, z, \theta)$, and precise alignment is required near the hole before sliding and rotating the object into place.

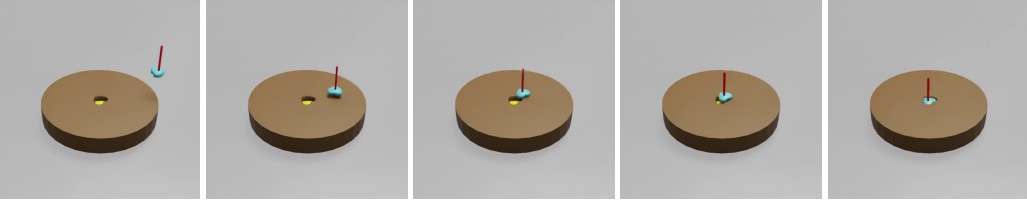

Figure 12: Example trajectory of Rigid Insertion task.

**Input and Output**    The input space for each node includes:

- Gripper nodes: `node_type`, position $\mathbf{p}_a$, velocity $\mathbf{v}_a$, angular velocity $\omega_a$.
- Object nodes: `node_type`, position $\mathbf{p}_o$, distance to target $d_{\text{target}}$.

The output consists of the gripper's linear velocity $v_a$ and a vector from which the angular velocity $\omega_a$ is derived. Specifically, the vector is decomposed into its parallel and tangential components with respect to the position vector $\mathbf{r}$, where $\mathbf{v}_\| = \left( \frac{\mathbf{v} \cdot \mathbf{r}}{\|\mathbf{r}\|^2} \right) \mathbf{r}$ and $\mathbf{v}_\perp = \mathbf{v} - \mathbf{v}_\|$. The angular velocity is then computed as $\omega_a = \frac{\mathbf{r} \times \mathbf{v}_\perp}{\|\mathbf{r}\|^2}$.

**Sample Space**

- Initial pose: $(x, y, z, \theta_{\text{yaw}}) \in [-1, 1]^2 \times [0, 0.5] \times [-\pi, \pi]$.
- Target pose: $\theta_{\text{yaw}} \in [-\pi, \pi]$.

**Reward Function**    The total reward consists of the following sub-rewards:

- **Distance to goal**:
$$R_{\text{goal}} = \|\mathbf{p}_o - \mathbf{p}_{goal}\|$$
  where $\mathbf{p}_o$ is the object's position, and $\mathbf{p}_{goal}$ is the target position.

- **Rotation distance**:
$$R_{\text{rotation}} = \texttt{quat\_diff}(\mathbf{r}_o, \mathbf{r}_{goal})$$
where $\mathbf{r}_o$ and $\mathbf{r}_{goal}$ are the object and goal orientations in quaternion.
- **Distance along $z$-axis**:
$$R_{\text{goal, z}} = \|\mathbf{p}_{o,z} - \mathbf{p}_{goal,z}\|$$
where $\mathbf{p}_{o,z}$ and $\mathbf{p}_{goal,z}$ represent the positions along the $z$-axis.

The total time-dependent reward with $T = 100$ is defined as:
$$R_{\text{tot}} = \begin{cases} -0.8R_{\text{goal}} - 2.0R_{\text{rotation}} - 0.4R_{\text{goal, z}}, & t < T - 2, \\ -4.0R_{\text{goal}} - 4.0R_{\text{rotation}} - 0.4R_{\text{goal, z}}, & t \geq T - 2. \end{cases}$$

## B.4 RIGID-INSERTION-TWO-AGENTS

The Rigid-Insertion-Two-Agents task extends Rigid-Insertion into 3D with two linear actuators controlling the object. The object's initial position and the target are sampled from the upper hemisphere. Control is limited to linear velocities along two axes, simplifying the task for stability in physical systems.

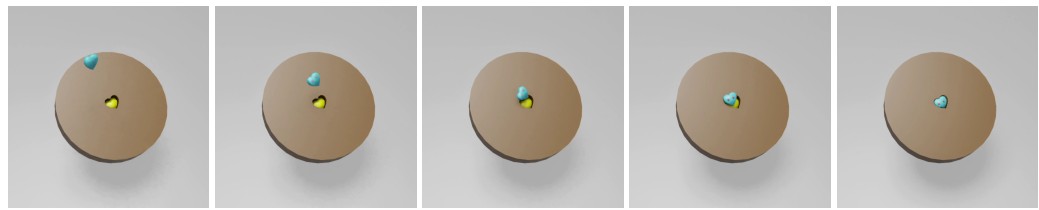

Figure 13: Example trajectory of Rigid Insertion with Two Agents task.

**Input and Output**    The input space for each node includes:

- Gripper nodes: `node_type`, position $\mathbf{p}_a$, velocity $\mathbf{v}_a$.
- Object nodes: `node_type`, position $\mathbf{p}_o$, distance to target $d_{\text{target}}$.

The output consists of the gripper's velocities; however, only the linear velocity $v_a$.

**Sample Space**

- Initial pose: translate in $x \in [0.25, 0.75]$, $y \in [-0.75, 0.75]$, $z \in [0.5, 1.25]$, and rotate around its own axis between $[-\pi, \pi]$.
- Target pose: $(\theta_{pich}, \theta_{yaw}) \in [-\pi/2, 0] \times [-\pi, \pi]$. These samples lie in an upper-hemisphere when rotating a unit-vector $[1, 0, 0]^T$ - an initial pose of the target placement, as shown in Figure 14.

**Reward Function**    The total reward consists of the following components:

- **Distance to goal**:
$$R_{\text{goal}} = \|\mathbf{p}_o - \mathbf{p}_{goal}\|$$
where $\mathbf{p}_o$ is the object's position, and $\mathbf{p}_{goal}$ is the target position.
- **Rotation distance**:
$$R_{\text{rotation}} = \texttt{quat\_diff}(\mathbf{r}_o, \mathbf{r}_{goal})$$
where $\mathbf{r}_o$ and $\mathbf{r}_{goal}$ are the object and goal orientations in quaternion.

The time-dependent reward with $T = 100$ is defined as:
$$R_{\text{tot}} = \begin{cases} -0.8R_{\text{goal}} - 0.08R_{\text{rotation}}, & t < T - 2, \\ -4.0R_{\text{goal}} - 0.6R_{\text{rotation}}, & t \geq T - 2. \end{cases}$$

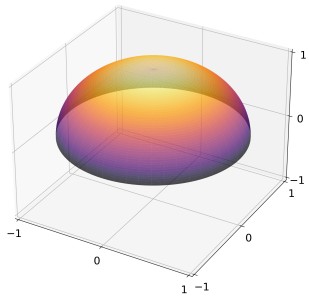

Figure 14: Sample space of the Rigid-Insertion-Two-Agents task.

## B.5 ROPE-CLOSING

In Rope-Closing, two actuators manipulate the endpoints of a deformable rope in a 2D plane, with the goal of wrapping the rope around a cylindrical object and closing the loop. The rope is segmented into 40 links, with each node representing the pose of a link.

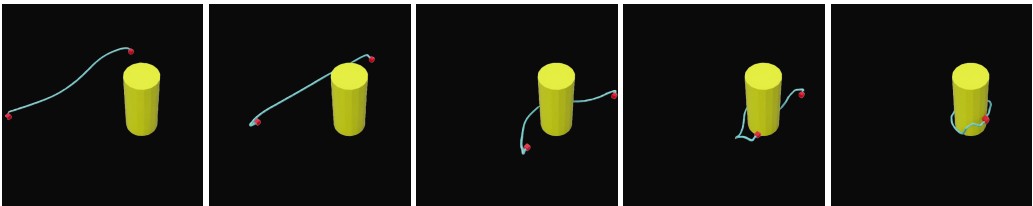

Figure 15: Example trajectory of Rope Closing task.

**Input and Output**   The input space for each node includes:

- Gripper nodes: node_type, position $\mathbf{p}_a$, velocity $\mathbf{v}_a$.
- Object nodes: node_type, position $\mathbf{p}_o$, velocity $\mathbf{v}_o$, distance to target $d_{\text{target}}$.

The output consists of the gripper's linear velocity $v_a$.

**Sample Space**   In this task, the rope starts in a straight configuration, and only the mid-point position is sampled, with the rope constrained to move accordingly,

- Initial rope mid-point: $\theta_{\text{yaw}} \in [-\pi/4, \pi/4]$.
- Target cylinder: $(x, y, \theta_{\text{yaw}}) \in [-0.5, 0.5]^2 \times [-\pi, \pi]$.

**Reward Function**   The total reward consists of:

- **Rope closing reward**:

$$R_{\text{rope closing}} = \|(x_{\text{gripper}_0}, y_{\text{gripper}_0}) - (x_{\text{gripper}_1}, y_{\text{gripper}_1})\|$$

- **Center wrapping reward**:

$$R_{\text{wrapping}} = \|(x_{\text{hanger}}, y_{\text{hanger}}) - (\mathbf{C}_{\text{rope},x}, \mathbf{C}_{\text{rope},y})\|$$

where $\mathbf{C}_{\text{rope}}$ is the center of the rope:

$$\mathbf{C}_{\text{rope}} = \frac{1}{n_{\text{nodes}}} \sum_{i=1}^{n_{\text{nodes}}} \mathbf{p}_{\text{node}_i}$$

- **Link velocity penalty**:

$$V_{\text{links}} = \frac{1}{n_{\text{links}}} \sum_{i=1}^{n_{\text{links}}} \|v_{\text{link}_i}\|$$

- **Action rate penalty**:

$$A_{\text{actions}} = \sqrt{a_i - a_{i-1}}$$

The total time-dependent reward with $T = 200$ is defined as:

$$R_{\text{tot}} = \begin{cases} -0.8R_{\text{wrapping}} - 0.01V_{\text{links}} - 0.001A_{\text{actions}}, & t < T - 20, \\ -2.0R_{\text{rope closing}} - 0.8R_{\text{wrapping}} - 0.01V_{\text{links}} - 0.001A_{\text{actions}}, & t \geq T - 20. \end{cases}$$

## B.6 ROPE-SHAPING

This task requires the rope to form a specific shape (e.g., a "W") to a desired orientation by controlling the actuators. The rope is segmented into 80 links, with each node representing the pose of a link.

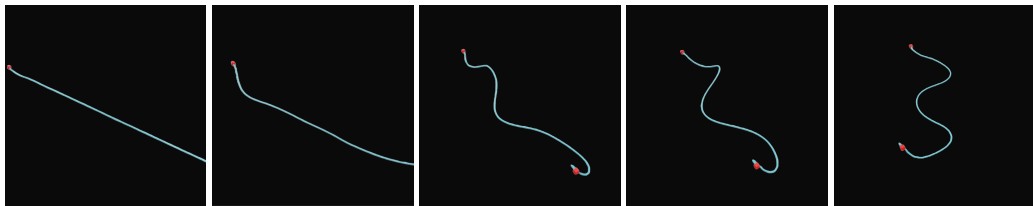

Figure 16: Example trajectory of Rope Shaping task.

**Input and Output**    The input space for each node includes:

- Gripper nodes: `node_type`, position $\mathbf{p}_a$, velocity $\mathbf{v}_a$.
- Object nodes: `node_type`, position $\mathbf{p}_o$, velocity $\mathbf{v}_o$, distance to target $d_{\text{target}}$.

The output consists of the gripper's linear velocity $v_a$.

**Sample Space**    In this task, the rope starts in a straight configuration, and only the mid-point position is sampled, with the rope constrained to move accordingly,

- Initial rope mid-point: $\theta_{\text{yaw}} \in [-\pi/2, -\pi/4] \cup [\pi/4, \pi/2]$.
- Target orientation: $\theta_{\text{yaw}} \in [-\pi/2, \pi/2]$.

**Reward Function**    To allow translation invariance, as the task focuses solely on shaping rather than the object's position, the shape descriptor is computed using both local and global geometric features of the rope. Given the positions of $N$ points along the rope, $\mathbf{p} = [\mathbf{p}_1, \ldots, \mathbf{p}_N]$, we first compute vectors between adjacent points, $\mathbf{v}_i = \mathbf{p}_{i+1} - \mathbf{p}_i$, and their normalized form $\hat{\mathbf{v}}_i = \mathbf{v}_i/\|\mathbf{v}_i\|$. The angles between consecutive segments are then $\theta_i = \arccos(\hat{\mathbf{v}}_i \cdot \hat{\mathbf{v}}_{i+1})$.

Next, we compute the global direction vector, $\mathbf{g} = \mathbf{p}_N - \mathbf{p}_1$, and normalize it as $\hat{\mathbf{g}} = \mathbf{g}/\|\mathbf{g}\|$. The angles between each segment and the global direction are $\theta_{\text{global},i} = \arccos(\hat{\mathbf{v}}_i \cdot \hat{\mathbf{g}})$.

Finally, we compute the relative positions of the points with respect to the midpoint $\mathbf{m} = (\mathbf{p}_1 + \mathbf{p}_N)/2$, and their distances $d_i = \|\mathbf{p}_i - \mathbf{m}\|$. The shape descriptor is formed by concatenating these angles and distances:

$$D_{\text{shape}} = [\theta_1, \ldots, \theta_{N-2}, \theta_{\text{global},1}, \ldots, \theta_{\text{global},N-1}, \mathbf{p}_1 - \mathbf{m}, \ldots, \mathbf{p}_N - \mathbf{m}, d_1, \ldots, d_N].$$

With the shape descriptor defined, the task's reward function encourages the current rope configuration to match the target shape, while also penalizing rapid changes in actions. The specific reward components are:

- **Shape matching reward**:

$$R_{\text{shape}} = \|D_{\text{current}} - D_{\text{target}}\|^2$$

where $D_{\text{current}}$ and $D_{\text{target}}$ represent the current and target shape descriptors.

- **Action rate penalty**:

$$A_{\text{actions}} = \sqrt{a_i - a_{i-1}}$$

where $a_i$ and $a_{i-1}$ represent the actions at consecutive time steps.

The total time-dependent reward with $T = 200$ is defined as:

$$R_{\text{tot}} = \begin{cases} -1.0R_{\text{shape}} - 0.0001A_{\text{actions}}, & t < T - 10, \\ -5.0R_{\text{shape}} - 0.0001A_{\text{actions}}, & t \geq T - 10. \end{cases}$$

## B.7 CLOTH-HANGING

In the Cloth-Hanging task, four actuators are attached to the corners of a cloth with a hole. The goal is to hang the cloth onto a beam by aligning the hole with the beam. The cloth is represented as a set of particles and simulated using a multi-body mass-spring-damper system. For the policy, only the hole boundary particles are used as the object representation (`knn` points around the hole centroid, with `knn_k = 10`). However, for the value function, all particle points are used, as required by the reward function defined below.

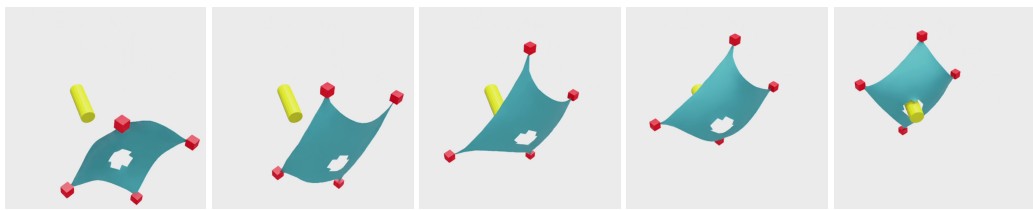

Figure 17: Example trajectory of Cloth Hanging task.

**Input and Output**  The input space for each node includes:

- Gripper nodes: `node_type`, position $\mathbf{p}_a$, velocity $\mathbf{v}_a$.
- Object nodes: `node_type`, position $\mathbf{p}_o$, velocity $\mathbf{v}_o$, distance to target hanger $d_{\text{target}}$, distance to initial shape $d_{\text{initial}}$. For the value function, instead of distances, the absolute coordinates of the target and initial node are used as features, while keeping the other features the same. We observed that providing the absolute coordinates of the target and initial points to each node makes learning easier. This may be due to the ability of $\text{MLP}_{\text{inner}}$ in DeepSets to infer similarity via dot products between feature channels.

The output consists of the grippers' linear velocity $v_a$.

**Sample Space**  In this task, the cloth always starts in a straight configuration, and the mid-point position of the cloth is sampled, with the cloth constrained to move accordingly:

- Hole location: each hole location $(x, y)$ is sampled within a predefined range of offsets from the cloth's center, with a fixed radius. In total, we generate 20 unique hole locations.
- Initial cloth mid-point: $\theta_{\text{pitch}} \in [-\pi, \pi]$.
- Target hanger: $(x, z) \in [-0.5, 0.5]^2$, and $(\theta_{\text{roll}}, \theta_{\text{pitch}}, \theta_{\text{yaw}}) \in [-\pi/4, \pi/2] \times [-\pi/2, \pi/2] \times [-\pi, \pi]$. This results in a sample that lies in the upper hemisphere with a quarter part of the lower hemisphere when rotating a unit-vector $[0, 1, 0]^T$ - an initial pose of the target hanger, as shown in Figure 18.

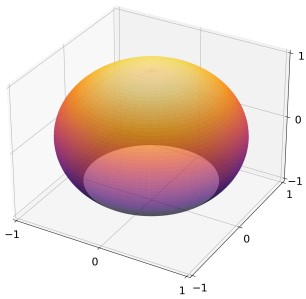

Figure 18: Sample space of the Cloth-Hanging task.

**Reward Function**  The total reward consists of the following sub-rewards:

- **Hole-hanger alignment reward**:

$$R_{\text{hole-hanger}} = \|\mathbf{c}_{\text{hole}} - \mathbf{c}_{\text{hanger}}\| + 0.1 \cdot |\cos(\theta_{\text{align}}) - 1|$$

where $\mathbf{c}_{\text{hole}}$ and $\mathbf{c}_{\text{hanger}}$ represent the centroids of the cloth's hole and the hanger, and $\theta_{\text{align}}$ measures their alignment. The first term is similar to the previous work from Antonova et al. (2021).

- **Point velocity penalty**:

$$V_{\text{points}} = \frac{1}{N} \sum_{i=1}^{N} v_{\text{point}_i}$$

where $v_{\text{point}_i}$ is the velocity of the $i$-th point on the cloth, and $N$ is the total number of points.

- **Cloth distortion penalty**:

$$D_{\text{distortion}} = \frac{1}{M} \sum_{i=1}^{M} \left| \frac{l_{\text{current}_i} - l_{\text{initial}_i}}{l_{\text{initial}_i}} \right|$$

where $l_{\text{current}_i}$ and $l_{\text{initial}_i}$ are the current and initial lengths of the $i$-th edge of the cloth, and $M$ is the total number of edges.

- **Action rate penalty**:

$$A_{\text{actions}} = \sqrt{(a_i - a_{i-1})^2}$$

where $a_i$ and $a_{i-1}$ represent the actions at consecutive time steps.

The total time-dependent reward with $T = 100$ is defined as:

$$R_{\text{tot}} = \begin{cases} -0.8R_{\text{hole-hanger}} - 0.2V_{\text{points}} - 1.0D_{\text{distortion}} - 0.002A_{\text{actions}}, & t < T - 2, \\ -4.0R_{\text{hole-hanger}} - 0.2V_{\text{points}} - 1.0D_{\text{distortion}} - 0.002A_{\text{actions}}, & t \geq T - 2. \end{cases}$$

## C  RIGID-BODY TASK OBJECTS

Table 1 presents the number of nodes used for each of the different geometrical shapes considered in the rigid manipulation experiments.

## D  FURTHER EXPERIMENTS

**Ablation on smaller sample space**  Here, we show a more detail version of the ablation in Figure 4. In addition to different sample spaces in the main paper, we evaluate all the methods on the scenarios when drawing $\theta_{\text{yaw}} \in (-\pi/8, \pi/8)$ and with a fixed-angle setting at $\theta_{\text{roll}} = 0, \theta_{\text{yaw}} = 0$.

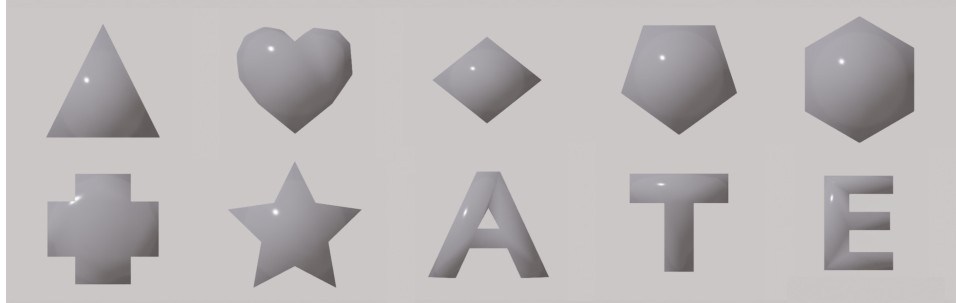

Figure 19: Overview of all objects used in the rigid manipulation tasks. From left to right: *Triangle*, *Heart*, *Diamond*, *Pentagon*, *Hexagon*, *Plus*, *Star*, *A-shape*, *T-shape*, and *E-shape*. In sliding tasks, all objects are used, while the *A-shape*, and *E-shape* are excluded in the insertion tasks due to its complex shape.

Table 1: Number of nodes for different geometrical shapes.

| Shape | Low Resolution #Nodes | High Resolution #Nodes |
|---|---|---|
| Triangle | 6 | 1128 |
| Diamond | 8 | 736 |
| Pentagon | 10 | 1032 |
| Hexagon | 12 | 1120 |
| T-shape | 16 | 1152 |
| Star | 20 | 1068 |
| Plus | 24 | 1224 |
| A-shape | 23 | 1660 |
| E-shape | 24 | 1972 |
| Heart | 25 | 1170 |

As shown in Figure 20, performance generally increases as the orientation range narrows. Interestingly, HeteroGNN performs better than its non-heterogeneous GNN in most cases, showing its high expressiveness though being less sample efficient. This phenomenon can be attributed to the shared networks among all the edge types of the naive GNN model. However, once employing EMPN as the backbone, HEPi consistently outperforms all the baselines in terms of both the performance and sample effeciency. This proves equivariant constraint plays a crucial role to reduce the problem complexity in large 3D space.

**Ablation on K-NN for obj-to-act edges**  Ablation in Figure 21 investigates the update of MPNN + VN$_{\text{Local}}$ with varying k-nearest neighbors on tasks *rigid-insertion*, *rigid-insertion-two-agents* and *rope-shaping*. For the two insertion tasks, the maximum node size is 25; therefore, we only vary $k$ from 1 to 10. On the other hand, *rope-shaping* task has 80 nodes, so $k$ is picked from $\{1, 10, 20, 50\}$. As shown, when $k$ is small, there are no-overlapping nodes, the actuator nodes can miss the information from distant nodes. Meanwhile, when $k$ is bigger, the number of overlapping nodes increase, and therefore, these nodes can be reached after one layer of message passing, thus resulting in higher expected return.

**Ablation on Number of Message-Passing Steps**  In this ablation, we examine the impact of varying the number of message-passing steps (#MPs) in both HEPi and a naive EMPN model. Our goal is to determine whether increasing the number of message-passing steps improves policy learning.

As shown in Figure 22, increasing the number of message-passing steps does not yield significant improvements. Moreover, Figure 23 specifically demonstrates that increasing number of message passing can lead to oversquashing, where the amount of information exponentially decays by the number of hops, and hence lead to performance drop. HEPi, on the other hand, efficiently reduces the information transmission time to only one hop. These results suggest that, in our design, fewer

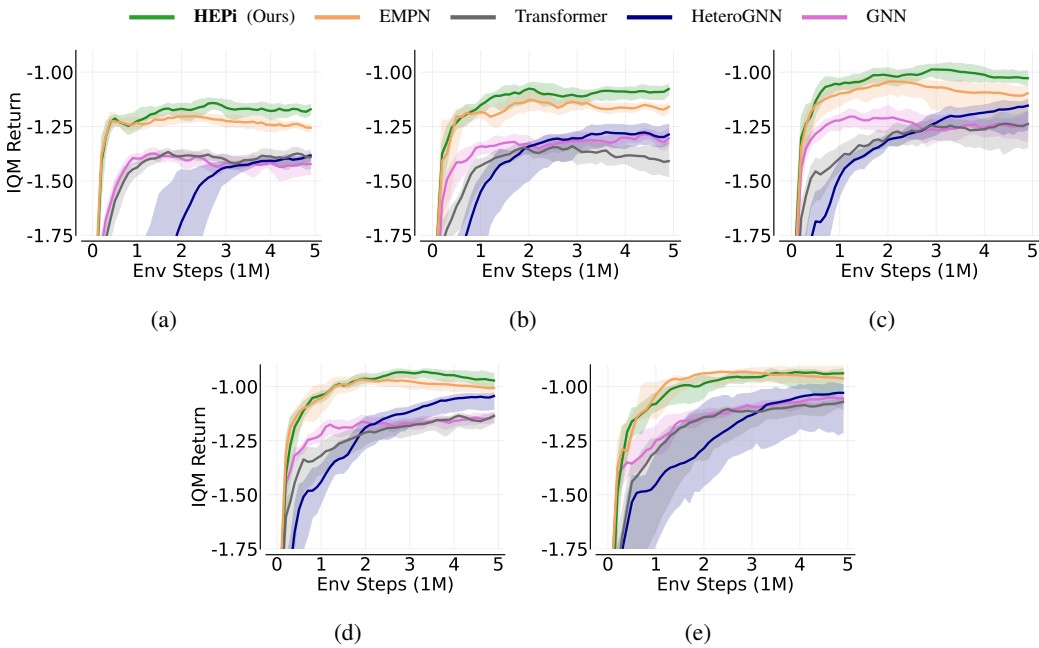

Figure 20: Performance of different models on the *Cloth-Hanging* task across various sample spaces. Assuming the global scene located at $r = [0, 1, 0]^T$, then from left to right, we generate sample by rotating $r$ by (a) $\theta_{\text{roll}} \in (-\pi/4, \pi/2)$, $\theta_{\text{yaw}} \in (-\pi, \pi)$, (b) $\theta_{\text{yaw}} \in (-\pi/2, \pi/2)$, and (c) $\theta_{\text{yaw}} \in (-\pi/4, \pi/4)$. Meanwhile, the bottom row shows results for (d) $\theta_{\text{yaw}} \in (-\pi/8, \pi/8)$, and (e) the fixed orientation at $\theta_{\text{roll}} = 0, \theta_{\text{yaw}} = 0$. As the sample space decreases, performance improves across all models, with HEPi consistently outperforming the baselines. The additional plot with fixed orientation on the bottom are averaged over 5 seeds while the others with 10 seeds.

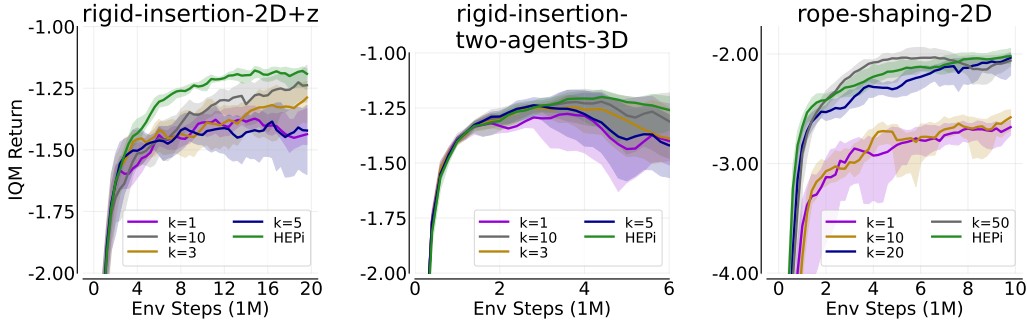

Figure 21: Ablation on different $k$-nearest neighbors choices for *obj-to-act* edges in MPNN + VN$_{\text{Local}}$ updates (in Section 3.3), evaluated across multiple tasks: *rigid-insertion*, *rigid-insertion-two-agents*, and *rope-shaping*. Results are averaged over 8 seeds.

message-passing steps suffice to capture the necessary information flow, reinforcing the efficiency of our graph design.

**Ablation on Orientation Discretization (ori_dim)** In this ablation, we explore the impact of varying the orientation discretization dimension (`ori_dim`) in the Equivariant Message Passing Network (EMPN). The `ori_dim` controls how finely we sample orientations from the $S^2$ sphere, where a higher `ori_dim` increases the number of samples and better approximates full equivariance. In our main experiments, we used a default `ori_dim` of 16. Here, we vary `ori_dim` across 8, 16, and 24.

As shown in Figure 24, increasing `ori_dim` generally improves performance in 3D tasks, as a finer discretization better captures orientation changes. However, this comes at the cost of increased

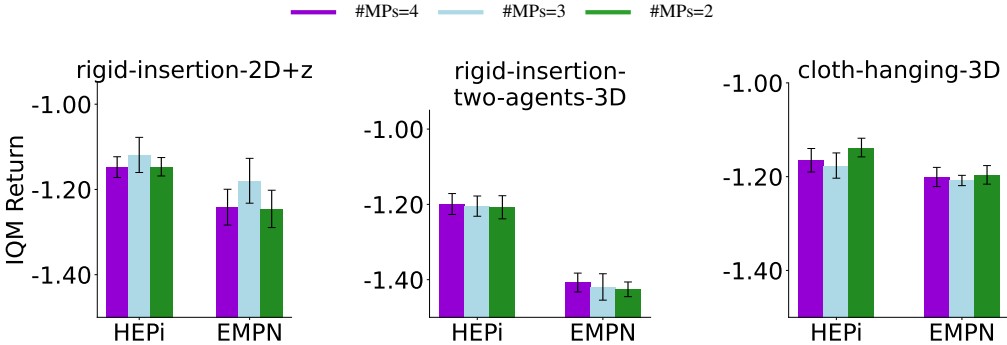

Figure 22: Ablation on the number of message-passing steps (#MPs) for HEPi and EMPN models. For HEPi, #MPs=$m$ refers to $m - 1$ object-to-object message-passing layers. Across all tasks, increasing the number of message-passing steps beyond a certain point does not improve performance, as the proposed graph design already efficiently transmits information from observations to actions. Results are averaged over 5 seeds.

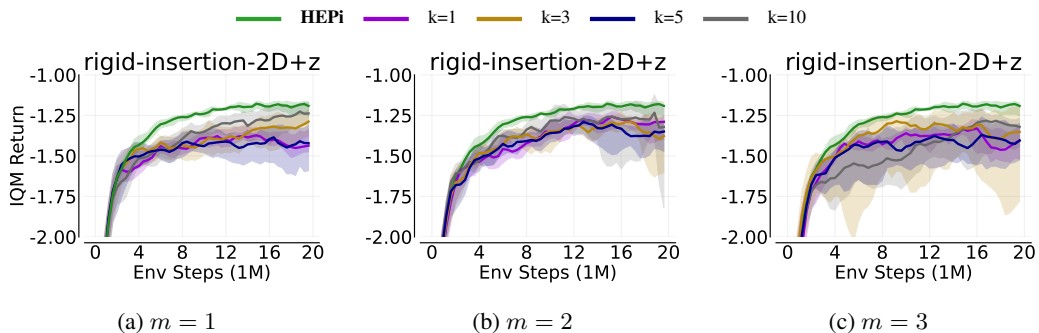

Figure 23: Ablation on different $k$-nearest neighbors for *obj-to-act* edges in MPNN + VN$_{\text{Local}}$ (in Section 3.3) updates, evaluated on the *Rigid-Insertion* task with varying message passing steps $m \in \{1, 2, 3\}$ for object nodes. Increasing the number of message passing steps degrades performance due to oversquashing. Results are averaged over 5 seeds.

training time due to the higher computational demand. Notably, in simpler 2D tasks, increasing `ori_dim` does not significantly affect performance, so we focus on 3D tasks for this ablation.

**Ablation on K-Nearest Neighbor Graph (KNN_k)** In this ablation, we evaluate the effect of varying the number of nearest neighbors (`KNN_k`) used to connect object nodes in our graph. Instead of relying on mesh edges, we use a K-nearest neighbor (KNN) graph to ensure a more generic representation, a common practice in PointCloud-based representations.

As shown in Figure 25, our default setting of `KNN_k=3` performs comparably to higher values of `KNN_k`. Increasing the number of nearest neighbors does not provide additional benefits and can even degrade performance slightly, as seen in the *rope-shaping-2D* task. This is likely due to the overcapacity of messages being passed through the network, which introduces unnecessary complexity in the message aggregation process.

## E EVALUATION DETAILS

### E.1 IMPLEMENTATION DETAILS

All experiments were conducted on a machine equipped with an NVIDIA A100 or an NVIDIA H100 GPU. We utilized the TorchRL framework (Bou et al., 2023) for the implementation of PPO and TRPL algorithms, and PyG (PyTorch Geometric) (Fey & Lenssen, 2019) for han-

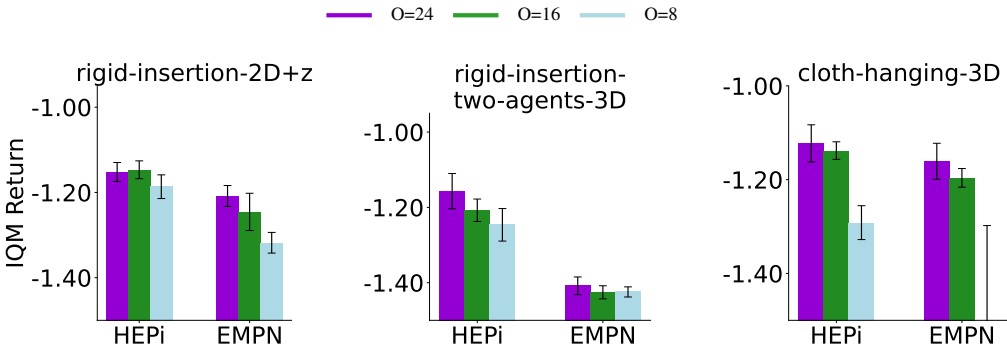

Figure 24: Ablation on the orientation discretization dimension (`ori_dim`). Increasing `ori_dim` improves performance in 3D tasks, such as *rigid-insertion* and *cloth-hanging*, by better approximating full equivariance. However, higher `ori_dim` also increases training time. Results are averaged over 5 seeds.

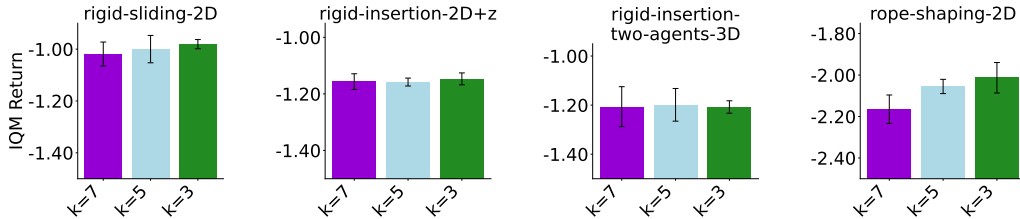

Figure 25: Ablation on the number of nearest neighbors (`KNN_k`) used in the KNN graph on HEPi. Increasing `KNN_k` beyond 3 does not improve performance and may even reduce it, particularly in tasks like *rope-shaping-2D*, due to message overcapacity. Results are averaged over 5 seeds.

dling the graph-based structure. The Transformer architecture was implemented using the `torch.nn.TransformerEncoder` and `torch.nn.TransformerEncoderLayer` packages from PyTorch (Paszke et al., 2017).

## E.2 Computational Time

We report the computational time for each model on all tasks here. Table 2 reports the total training time and Table 3 demonstrates the size of the input graph, which is the main factor contributing to the total training time.

Table 2: Total training time for each task (in hours). (*) We note that the fast training speed of Transformer might be attributed to the internal optimization implementation of `PyTorch`. (**) In *Rope-Shaping* task, we report the training time on NVIDIA H100 GPU.

|  | HEPi | EMPN | Transformer* |
|---|---|---|---|
| Rigid-Sliding | 2h 10m | 2h 56m | 1h 15m |
| Rigid-Pushing | 2h 58m | 3h 58m | 1h 52m |
| Rigid-Insertion | 1h 56m | 2h 35m | 1h 14m |
| Rigid-Insertion-Two-Agents | 1h 3m | 1h 20m | 36m |
| Rope-Closing | 1h 14m | 1h 40m | 51m |
| Rope-Shaping** | 2h 58m | 4h 57m | 1h 56m |
| Cloth-Hanging | 2h 40m | 2h 36m | 2h 21m |

Table 3: Maximum number of nodes and type of connection for each task.

| Shape | Maximum #nodes | Graph Connections |
|---|---|---|
| Rigid-Sliding | 25 | knn=3 |
| Rigid-Insertion | 25 | knn=3 |
| Rigid-Pushing | 25 | knn=3 |
| Rigid-Insertion-Two-Agents | 25 | knn=3 |
| Rope-Closing | 40 | knn=3 |
| Rope-Shaping | 80 | knn=3 |
| Cloth-Hanging | 10 | complete |

### E.3 GRID SEARCH FOR PPO

To fairly compare PPO with TRPL, we perform a grid search for PPO over 5 seeds and select the best-performing configuration based on the maximum return. We then run the chosen configuration on 5 additional seeds and report the results in Figure 8. Specifically, we tune the `clip_eps` parameter, which controls how much the new policy is allowed to deviate from the old policy, with values {0.1, 0.2, 0.3, 0.5}, and explore both with and without annealing (`anneal_clip_eps=True or False`). The `clip_eps` bounds the probability ratio between the new and old policies to $(1 - \epsilon, 1 + \epsilon)$, preventing large updates and ensuring stable learning. Lower values of `clip_eps` result in more conservative updates, while higher values allow more flexibility in policy updates. Annealing progressively decreases `clip_eps` over time, tightening the constraint as training progresses.

### E.4 HYPERPARAMETERS

We presents the hyperparameters used across all policy models (HEPi, EMPN, and Transformer) for all the tasks in Table 4.

Table 4: Hyperparameters used for all tasks. In EMPN, the number of layers (*) corresponds to the number of message-passing steps.

| | HEPi | EMPN | Transformer |
|---|---|---|---|
| contextual std | true | true | true |
| latent dim. | 64 | 64 | 64 |
| activation | GELU | GELU | ReLU |
| dropout | false | false | false |
| num layers | n.a. | 2* | 2 |
| num heads | n.a. | n.a. | 2 |
| num messages (obj-to-obj) | 1 | n.a. | n.a. |
| num messages (obj-to-act) | 1 | n.a. | n.a. |
| num messages (act-to-act) | 1 | n.a. | n.a. |
| ponita orientation dim. | 16 | 16 | n.a. |
| ponita degree | 2 | 2 | n.a. |
| ponita spatial hidden dim. | [64, 64] | [64, 64] | n.a. |
| ponita fiber hidden dim. | [64, 64] | [64, 64] | n.a. |
| ponita widening factor | 4 | 4 | n.a. |

The following tables, Table 5 and Table 6 provide details on environment settings, data collection parameters, and training hyperparameters.

Table 5: Hyperparameters for Rigid Environments

| | Rigid-Sliding | Rigid-Insertion / Rigid-Pushing | Rigid-Insertion -Two-Agents |
|---|---|---|---|
| **Environments** | | | |
| time steps | 100 | 100 | 100 |
| warmup steps | 5 | 5 | 5 |
| episode length (in sec.) | 4 | 4 | 4 |
| decimation | 4 | 4 | 4 |
| simulation $dt$ | 0.01 | 0.01 | 0.01 |
| **Data Collection** | | | |
| frames per batch | 100k | 100k | 100k |
| total frames | 20M | 20M / 30M | 6M |
| **Input Graph** | | | |
| obj-to-obj edges | knn=3 | knn=3 | knn=3 |
| act-to-act edges | n.a. | n.a. | complete |
| **Training** | | | |
| epochs | 5 | 5 | 5 |
| mini-batch size | 1000 | 1000 | 1000 |
| learning rate (actor) | 3e-4 | 3e-4 | 3e-4 |
| learning rate (critic) | 3e-4 | 3e-4 | 3e-4 |
| critic coeff. | 0.5 | 0.5 | 0.5 |
| entropy coeff. | 0.005 | 0.005 | 0.005 |
| clip gradient norm | false | false | false |
| **Projection** | | | |
| trust region coeff. | 4.0 | 1.0 | 1.0 |
| mean bound | 0.05 | 0.05 | 0.05 |
| covariant bound | 0.001 | 0.0025 | 0.0025 |
| **Critic (DeepSets)** | | | |
| num inner layers | 2 | 2 | 2 |
| num outer layers | 2 | 2 | 2 |
| hidden dim. | 64 | 64 | 64 |
| activation | ReLU | ReLU | ReLU |
| layer norm | true | true | true |

Table 6: Hyperparameters for Deformable Environments

|  | Rope-Closing | Rope-Shaping | Cloth-Hanging |
|---|---|---|---|
| **Environments** | | | |
| time steps | 200 | 200 | 100 |
| warmup steps | 10 | 10 | 10 |
| episode length (in sec.) | 4 | 4 | 2 |
| decimation | 2 | 2 | 2 |
| simulation $dt$ | 0.01 | 0.01 | 0.01 |
| **Data Collection** | | | |
| frames per batch | 40k | 40k | 10k |
| total frames | 4M | 10M | 5M |
| **Input Graph** | | | |
| obj-to-obj edges | knn=3 | knn=3 | complete |
| act-to-act edges | complete | complete | complete |
| **Training** | | | |
| epochs | 5 | 5 | 5 |
| mini-batch size | 200 | 200 | 200 |
| learning rate (actor) | 3e-4 | 3e-4 | 3e-4 |
| learning rate (critic) | 3e-4 | 3e-4 | 3e-4 |
| critic coeff. | 0.5 | 0.5 | 0.5 |
| entropy coeff. | 0.005 | 0.005 | 0.005 |
| clip gradient norm | false | false | false |
| **Projection** | | | |
| trust region coeff. | 4.0 | 1.0 | 4.0 |
| mean bound | 0.05 | 0.05 | 0.05 |
| covariant bound | 0.001 | 0.0025 | 0.001 |
| **Critic (DeepSets)** | | | |
| num inner layers | 2 | 2 | 2 |
| num outer layers | 2 | 2 | 2 |
| hidden dim. | 64 | 64 | 64 |
| activation | ReLU | ReLU | ReLU |
| layer norm | true | true | true |

