# OpenReview forum: "Geometry-aware RL for Manipulation of Varying Shapes and Deformable Objects"
_ICLR.cc/2025/Conference — ICLR 2025 Oral_

### Official Review · Reviewer_6nhr · 2024-10-28

**Soundness:** 3
**Presentation:** 3
**Contribution:** 2
**Rating:** 8
**Confidence:** 2

**Summary:**

The paper trains policies for one or two end effectors to manipulate rigid and deformable objects (ropes and cloths). It models the actuator and object of interest with a heterogeneous graph and advocates for a heterogeneous equivariant policy (HEPi). The paper benchmarks its method on tasks including manipulating rigid and deformable objects in IsaacLab and reports its superior performance over typical baselines.

**Strengths:**

In general, I find the story in this paper intuitive and convincing. While the two core building blocks (heterogeneous graphs and equivariant policies) have been explored in prior arts, this paper presents a proper combination of both in the setting of rigid/deformable object manipulation. I feel the proposed method has the potential to be extended to manipulate fluid and rigid/soft/fluid-coupled systems, and it would be a pleasant surprise if the paper could demonstrate a fluid scene similar to the “Pour Water” task in Lin et al 2020. Of course, I understand this is outside the scope of the current submission.

**Weaknesses:**

I generally agree with the limitations listed in the paper. I think the paper can be improved in the following ways:

1. While the technical method and the story look promising, the benchmark scenes are still relatively simple because the setup only models a point-like end effector, ignoring kinematic and dynamic constraints in practical robotic hand/arm structures. Therefore, transferring the result to a real-world robot is not straightforward and probably requires more algorithmic development.

2. From a dynamic perspective, the 2D rigid-sliding task is too trivial to be a valuable benchmark for evaluating this paper and its baselines. This is already reflected in the result: “HEPi and Transformer policies perform comparably, suggesting that the limited task complexity does not fully leverage the benefits of equivariant constraints. ” I suggest this experiment remove the suction gripper and try pushing the rigid object to its target position and orientation with one/two end effectors and their contact with the object, similar to the setup in http://sain.csail.mit.edu/.

3. Deformable objects can be classified by their dimensions: codimension-2 (e.g., ropes), codimension-1 (e.g., cloths), and codimension-0 (e.g., rubber balls). The paper seems to consider the first two categories only. Handling full-dimension soft objects in the proposed method may be non-trivial because it requires observations of object coordinates and cannot be easily obtained by “integrating state-of-the-art computer vision techniques to extract key points from cameras.” I suggest the paper clarify this point by being more precise with the concept of deformable objects it deals with.

**Questions:**

Please see the weaknesses above.

---

> ### Author Response · Authors · 2024-11-22
>
> We thank the reviewer for their time and effort and for acknowledging that we present a proper combination of recent advances for rigid and deformable object manipulation.
>
> > Evaluate on fluid scene similar to the “Pour Water” task in Lin et al 2020
> >
> - We thank the reviewer for their insightful observation about the potential applicability of our approach to fluid manipulation tasks. We agree that HEPi's design could be extended to such scenarios by constructing a k-nearest-neighbor graph from fluid particles, making it well-suited for dealing with fluid manipulation tasks. However, due to time constraints, we focused on introducing the new *Rigid-Pushing* task in this revision, as we believe it aligns more closely with our story and could strengthen our contributions. We look forward to exploring fluid manipulation as a future research direction.
>
> > point-like end effector setting and rigid-pushing task recommendation
> >
> - We thank the reviewer for the suggestion regarding the pushing task with a concrete example. Based on this feedback, we introduced a new Rigid-Pushing task in this revision, where the actuator must push the object without a direct connection, increasing the task's dynamical complexity. In addition, we conducted a noise sensitivity analysis to evaluate HEPi's robustness to input noise and scalability to high-resolution objects. These additions further highlight HEPi’s applicability to real-world scenarios, addressing common challenges like sensory noise and diverse object representations.
> - Next, we acknowledge the reviewer’s concern about the use of point-mass actuators, which simplifies the kinematic and dynamic constraints of practical robotic structures. However, in this paper, we concentrate on task-space exploration via end-effector control. This helps to isolate the control and learning problem in manipulation, highlighting the challenges of geometrical understanding in policy learning we want to address.
>
> > rigid-sliding is too trivial
> >
> - Regarding the Rigid-Sliding task, we agree it might seem trivial at first glance from a dynamical perspective. However, we believe it serves as an essential first step in the benchmark, showcasing the capability of handling multiple geometries in simpler settings. This progression helps demonstrate the importance of design choices: for instance, moving from Rigid-Sliding to Rigid-Insertion, where aligning the object with additional z-axis movement adds complexity, highlights the value of heterogeneity. Such tasks allow us to systematically evaluate each component of the model and its effectiveness in capturing task-specific challenges.
>
> > Handling full-dimension soft objects in the proposed method may be non-trivial.
> >
> - Thank you for your insightful comment, and we apologize for the unclear phrasing regarding our method. We would like to clarify that our approach does not require full object coordinates but only keypoint coordinates. These coordinates are used to construct a k-nearest neighbors (kNN) graph for the object subgraph. Extracting such keypoints can be achieved using state-of-the-art computer vision techniques, such as [1, 2], which effectively detect keypoints from visual inputs. We also revised this point in the manuscript.
>
> [1] Hou, C. et al. Key-Grid: Unsupervised 3D keypoints detection using grid heatmap features. *Advances in Neural Information Processing Systems (NeurIPS)*, 2024.
>
> [2] Tumanyan, N. et al. DINO-Tracker: Taming DINO for self-supervised point tracking in a single video. *European Conference on Computer Vision (ECCV)*, 2024.

---

> > ### Comment · Reviewer_6nhr · 2024-11-25
> > **Thank you for the response**
> >
> > Thank you for the revision. It looks like the new Rigid-Pushing task adds a 3D rod to interact with an object that stays on a 2D plane. I do not think this is substantially more difficult than Rigid-Sliding, but I am OK with it.
> >
> > Regarding full-dimension soft objects: I meant that solving a 3D volumetric deformable solid's motion generally requires knowledge of its deformation over the whole body, including information about how its interior deforms. Such information is not accessible from a 2D image of the object because such an image only shows the surface, not the interior, of the object.

---

> > > ### Author Response · Authors · 2024-11-25
> > >
> > > Thank you for your response. However, we have a different opinion on the new Rigid-Pushing task. From a dynamic perspective, unlike Rigid-Sliding, Rigid-Pushing requires substantially more steps to complete the task. Specifically, the rod, controlled via linear velocity without angular velocity, must first approach and make contact with the object, then continuously push and reorient it to match the target configuration. Additionally, as shown in the revised Figure 3, HEPi significantly outperforms both EMPN and Transformer baselines, highlighting the importance of explicit heterogeneity modelling and equivariance for this task, even though the movement remains constrained to a 2D plane, as in Rigid-Sliding.
> > >
> > > Regarding the second point, we thank you for the clarification, which has helped us better understand HEPi's limitations. We agree that under limited sensing scenarios, HEPi may struggle to solve tasks that require capturing the internal state of volumetric deformable objects, as you have noted.

---

> > > > ### Comment · Reviewer_6nhr · 2024-12-03
> > > > **Thank you for the update**
> > > >
> > > > After reading the rebuttal and other reviews, I am happy to support the acceptance of this work. A score of 7 would reflect my true feelings about this work. As 7 is not an option in the system, I will raise my score to 8 but decrease my confidence to 2 instead.

---

### Official Review · Reviewer_ozCf · 2024-11-02

**Soundness:** 3
**Presentation:** 3
**Contribution:** 3
**Rating:** 8
**Confidence:** 3

**Summary:**

This paper addresses the task of manipulating diverse rigid and deformable objects with geometry-aware rl policy.

A heterogeneous graph is proposed to represent rigid and deformable object manipulation tasks. To leverage geometric symmetry for better task performance and sample efficiency, a heterogeneous equivariant policy that utilizes SE(3) equivariant message passing networks is proposed.

Evaluation is carried out on a self-curated rl benchmark, including rigid insertion of diverse objects, as well as rope and cloth manipulation with multiple end-effectors. The proposed method outperforms baseline methods in terms of average returns, sample efficiency, and generalizability.

**Strengths:**

The proposed heterogeneous graph representation is a general representation for both rigid and deformable object manipulation tasks, and captures the geometric structure of the object. HEPi is a novel formulation of graph-based equivariant policy build on top of the heterogeneous graph representation that enables rl for manipulation of diverse shapes and deformable objects.

The paper has demonstrated thorough evaluations and ablations in simulation, with convincing results proving that the proposed method has better performance, sample-efficiency, and generalizability. Detailed discussions are provided for the results, providing interesting insights.

**Weaknesses:**

The object model that contains vertices are required for the object node representation, which is the main observation for the policy. This might not be easy to get in the real world due to a lot of self-occlusions during deformable object manipulation.

The paper demonstrates thorough evaluations across various simulation benchmarks, but lacks real-world evaluations to make the method fully convincing in terms of it's usefulness to real-world robot applications.

typo at line 74: equivariacne->equivariance.

**Questions:**

Could the authors expand more on line 51-52 about attention making the optimization landscape more difficult to traverse?

Would the policy be able to generalize to objects with different physical properties?

---

> ### Author Response · Authors · 2024-11-22
>
> We thank the reviewer for their time and effort and for acknowledging that we present a novel and general formulation for challenging manipulation tasks and demonstrate its effectiveness through a thorough evaluation.
>
> Based on the reviewers’ input we added additional experiments as detailed in our general answer.
>
> > Self-Occlusion and Sim2Real Transfer
> >
> - We acknowledge the challenge of self-occlusions during deformable object manipulation and the difficulty of obtaining object node representations in real-world settings. Fundamentally, addressing this issue would require a PoMDP formulation and including some form of history, which is beyond the scope of this paper. However, we believe it is a promising future research direction. Next, as also pointed out by other reviewers, sim2real transfer might not be directly applied. To this end, we introduced a new experiment on the *Rigid-Pushing* task and conducted an analysis of HEPi’s robustness to noisy inputs and scalability to high-resolution objects. We refer the reviewer to our detailed discussion in the response to Reviewer **qL9N** for more information on these analyses.
>
> > Generalization to Physical Properties
> >
> - One reason we chose NVIDIA IsaacLab is its demonstrated success in sim2real transfer for locomotion tasks via massively parallel RL training and domain randomization techniques [1, 2]. We believe that HEPi could benefit from this idea and believe this is an interesting research direction for future work.
>
> > Attention makes pptimization landscape more difficult to traverse
> >
> - Regarding the reviewer’s question about attention mechanisms, where in the manuscript, we argued that adding attention often introduces additional parameters, making the optimization landscape more challenging to traverse. To better clarify this point, we would like to stress that unlike supervised learning, on-policy reinforcement learning relies on high-frequency data collection and efficient adaptation, which can be hindered by large, overparameterized models, as shown in the Appendix of [3] (C49 - Fig. 18, C52 - Fig. 22). Our lightweight heterogeneous equivariant architecture is specifically designed to mitigate these challenges, enabling efficient on-policy training while preserving expressiveness.
>
> [1] Rudin, N. et al. Learning to walk in minutes using massively parallel deep reinforcement learning. Proceedings of the 5th Conference on Robot Learning (CoRL), PMLR 164:91–100, 2022.
>
> [2] Mittal, M. et al. Orbit: A unified simulation framework for interactive robot learning environments. IEEE Robotics and Automation Letters (RA-L), 8(6):3740–3747, 2023.
>
> [3] Andrychowicz, M. et al. What matters for on-policy deep actor-critic methods? A large-scale study. International Conference on Learning Representations (ICLR), 2021.

---

### Official Review · Reviewer_Z2z8 · 2024-11-03

**Soundness:** 3
**Presentation:** 4
**Contribution:** 3
**Rating:** 8
**Confidence:** 4

**Summary:**

The authors propose a novel SE(3) equivariant RL method called HEPi, and introduce a new benchmark for future geometry-aware RL evaluations.  The paper use EMPNs as algorithm backbone to allow model generalize between poses and its heterogeneity. The paper use graph as representation and do thorough mathmetical analyze.

**Strengths:**

+ Clever Design: This paper includes several ingenious designs.
   + The use of EMPNs The use of SE(3)-equivariant networks ensures that the model naturally possesses SE(3) generalization, thereby reducing the search space and complexity.
   + The use of TRPL the introduction of TRPL in place of traditional PPO makes hyperparameter tuning easier, addressing a significant challenge in RL training.

+ Detailed Appendix Experimental Description: One of the major issues in RL is poor reproducibility. However, in this paper, the authors provide a detailed appendix that lists experimental information, including the reward function and hyperparameters. This makes the paper highly reproducible.

+ Benchmark Design: This paper provides a detailed demonstration of the proposed benchmark tasks in the video and clearly defines these tasks in the appendix. As a result, these tasks and environments can be easily adopted by the research community.

I am inclined to accept this paper.

**Weaknesses:**

+ Real-World Application: As the authors note, this paper uses ground-truth model inputs and does not incorporate a physical robot, which means it cannot be directly applied to real-world scenarios. However, given the significant contributions of this paper in terms of benchmarking and methodology, I do not consider this a critical issue. Nonetheless, I still recommend that the authors include some specific experiments to evaluate this aspect. I suggest author can simply add visual capture pipeline. The details are as follows:
    + For task like rigid insertion and rigid sliding, you can simply add camera to take picture then use a common pose estimation module to estimate the position. The potential errors in pose estimation can be used to test the robustness of the proposed method to input errors.
    + For tasks involving fabrics or ropes, you can directly use a camera to capture point clouds and then construct a graph using the point cloud data.
+ Geometry aware: This paper does not seem to have any special designs focused on geometry, although graphs do play a role. However, the primary focus of the paper appears to be on SE(3)-equivariant designs.

**Questions:**

Thank you for the detailed appendix, which has addressed my questions about the specifics. If the authors can supplement the experiments as suggested, I would be willing to increase my score.

---

> ### Author Response · Authors · 2024-11-22
>
> We thank the reviewer for their time and effort and for acknowledging that we present a clever design and highly reproducible method and benchmarks. Based on your, and the other reviewers’ feedback, we extended our evaluation to analyze the applicability of our methods in real-world scenarios:
>
> We appreciate the reviewer’s suggestion regarding real-world applicability and visual pipelines. In response to this concern, we have taken the following steps, detailed in our reply to Reviewer **qL9N** and summarized here:
>
> - We introduced a new Rigid-Pushing task, specifically designed to test HEPi's scalability and robustness.
> - We evaluated HEPi on this task using high-resolution objects and analyzed its performance under varying levels of Gaussian noise. These experiments simulate noisy sensory inputs, closely mimicking real-world conditions.
>
> As shown in the new Figure 5, HEPi demonstrates strong resilience to noisy inputs and scales effectively to high-resolution objects.
>
> While we acknowledge the importance of incorporating a full vision pipeline, time constraints limited our ability to implement and integrate this into the current work. Instead, we opted for a controlled and systematic analysis, simulating noisy sensory inputs to closely mimic real-world conditions. We believe this approach provides meaningful insights into HEPi's robustness and scalability while serving as a foundation for future extensions to real-world settings. We hope these additional analyses can address your concerns.
>
> Regarding the “geometry awareness”: Following the prior work on Geometric Deep Learning [1], we use the term “geometry-aware” to describe our approach to modeling reinforcement learning for manipulation tasks as a geometric graph problem, with actuator and object nodes situated in a Euclidean space. Furthermore, our HEPi framework is built upon the $SE(n)$ Equivariant Message Passing Network [2], which is specifically designed to respect the symmetries and geometric properties inherent in the data. However, we acknowledge the term “geometry” has different meanings across research communities and if any part of our explanation remains unclear, we welcome suggestions from the reviewer to help further improve the manuscript.
>
> [1] Bronstein, M. M. et al. Geometric deep learning: Grids, groups, graphs, geodesics, and gauges. *arXiv preprint arXiv:2104.13478*, 2021.
>
> [2] Bekkers, E. et al. Fast, Expressive $SE(n)$ Equivariant Networks through Weight-Sharing in Position-Orientation Space. International Conference on Learning Representations (ICLR), 2024.

---

> > ### Comment · Reviewer_Z2z8 · 2024-12-02
> >
> > Thank you for your response. I acknowledge the effort you have put in during the rebuttal period. Additionally, if you could incorporate a full vision pipeline into the final version, this would be an excellent paper. I am raising my score from 6 to 8 and am inclined towards accepting this article.

---

> > > ### Author Response · Authors · 2024-12-02
> > >
> > > Thank you again for your thoughtful feedback and for considering raising your score. We are looking into the possibility of incorporating a vision pipeline into our tasks, as suggested.
> > >
> > > However, it seems your original rating remains unchanged. We’re sorry for disturbing you, but if you could take a moment to update it, we would greatly appreciate it.

---

### Official Review · Reviewer_qL9N · 2024-11-04

**Soundness:** 4
**Presentation:** 3
**Contribution:** 3
**Rating:** 8
**Confidence:** 3

**Summary:**

The paper proposes a novel setting for representing robotic manipulation problems as heterogeneous graph learning problems. The authors introduce a graph-based policy model, *Heterogeneous Equivariant Policy (HEPi)*, featuring multiple *SE(3)* equivariant message-passing networks (EMPNs) to model smaller sub-graphs like actuators and objects. HEPi explicitly models heterogeneity by assigning distinct network parameters for each interaction type to reduce message mixing and improve expressiveness. HEPi is claimed to be the first study of equivariant policies on 3D space within a reinforcement learning setting for robotic manipulation.

The authors theoretically prove that, for HEPi any two actuator and object nodes can exchange information while the graph network with locally connected actuators and object nodes can not. This justifies their design of actuator nodes as global virtual nodes to connect all object nodes

The authors test the proposed heterogeneous graph representation with reinforcement learning for 6 rigid and deformable object tasks, including Rigid-Sliding, Rigid-Insertion, Rigid-Insertion-Two-Agents, Rope-Closing, Rope-Shaping, Cloth-Hanging. Empirical results show that the proposed approach is more sample efficient and less likely to converge on sub-optimal solutions compared with the state-of-the-art Transformer and non-heterogeneous EMPN methods.

**Strengths:**

1. The paper is well-written and presents its ideas clearly with solid formulation.
2. The proposed approach of modeling robotic manipulation problems as heterogeneous graph learning problems is well-motivated and clever to unify the structure for both rigid and deformable object tasks using sub-graphs for actuators and objects.
3. The adaptation of *SE(3)* equivariant message passing networks is reasonable and suitable to exploit the geometric symmetry for improving the sample efficiency in the large 3D search space of configurations. As far as I know, HEPi is one of the first studies of equivariant policies on 3D space within a reinforcement learning setting for robotic manipulation.
4. The empirical experiments are comprehensive and sound to support the arguments. Most results are averaged over 10 seeds using
interquartile mean with 95% confidence intervals, which is also statistically robust.

**Weaknesses:**

1. The experiments are limited to simple geometric shapes like ropes, triangles, and stars, which can be easily modeled using a few nodes (< 100 in all the experiments). However, the target objects in most 3D manipulation tasks are more complex and sophisticated, e.g., cabinets or dresses, which require significantly more nodes to model. I doubt whether HEPi can still generalize to all those complicated geometries, which brings much computational complexity to graph learning.
2. The current framework assumes that the object coordinates are readily available in the observation, which is basically a simulation setting. In real applications without coordinates, such coordinates can only be extracted from other CV models in the wild. As a graph policy model, HEPi is likely to suffer from the cumulated error in the observation and fail the tasks compared to non graph learning methods.
3. Some typos:
- L17: objects -> object.
- L182: “lifts”, the left quotation mark.
- L344: There should be a blank space before HEPi.

Nevertheless, the overall idea is novel, and empirical experiments are solid enough to support the proposed setting. I recommend that the paper should be accepted.

**Questions:**

1. Can HEPi generalize to more complicated geometries in the real world other than simple ropes or shapes like hearts and stars, which might bring much computational complexity to graph learning?
2. How sensitive is HEPi to the perturbation of the object coordinates from the observation?

---

> ### Author Response · Authors · 2024-11-22
>
> We thank the reviewer for their time and effort and for acknowledging that we propose a well-motivated and clever approach whose effectiveness is supported by a comprehensive and sound set of experiments.
>
> We appreciate the pointed-out typos, as well as the feedback and questions about our experiments. As detailed in our general answer, we added additional experiments in the revised submission, the Rigid-Pushing task, to answer:
>
> - **Scalability to high-resolution objects:** One key property of graph neural networks is their ability to scale to higher resolution in zero-shot fashion since GNNs are designed to capture local information via message passing mechanisms [1]. We evaluate this on the newly designed rigid-pushing task. The training setup mirrors that of other rigid tasks, with 10 objects of varying geometries (average ~20 nodes). During evaluation, we tested HEPi on finer objects with significantly higher resolution (average ~1200 nodes).
> - **Sensitivity to perturbations:** To answer the question, “How sensitive is HEPi to the perturbation?”, we also report the average returns in Figure 5 using the best checkpoint on varying noise scales on both datasets with low and high resolution. As shown, HEPi maintains high returns even under significant observation noise.
>
> To summarize, the results in Figure 5 demonstrate that HEPi generalizes effectively to higher-resolution inputs and maintains strong performance under noisy observations, showing robustness to sensory inaccuracies commonly encountered in real-world deployments.
>
> We now want to address the remaining open points:
>
> - Regarding the need for object coordinates, our approach does not rely on full object meshes, but instead uses keypoint coordinates to construct the k-nearest neighbors (kNN) graph for the object subgraph. Such keypoints can be extracted using advanced computer vision techniques, e.g., in [2, 3]. For tasks requiring observable object velocities, such as RigidPushing, RopeShaping, and ClothHanging; these can be measured or estimated using historical data derived from sequential keypoint observations. While we acknowledge that this introduces additional challenges, addressing this problem is outside the scope of the current paper.
>
> [1] Li, Z. et al. Multipole graph neural operator for parametric partial differential equations. *Advances in Neural Information Processing Systems (NeurIPS)*, 2020.
>
> [2] Hou, C. et al. Key-Grid: Unsupervised 3D keypoints detection using grid heatmap features. *Advances in Neural Information Processing Systems (NeurIPS)*, 2024.
>
> [3] Tumanyan, N. et al. DINO-Tracker: Taming DINO for self-supervised point tracking in a single video. *European Conference on Computer Vision (ECCV)*, 2024.

---

> > ### Comment · Reviewer_qL9N · 2024-12-02
> >
> > Thanks for clarifying the concerns. It is an interesting read.

---

### Author Response · Authors · 2024-11-22
**Answer to all reviewers**

We thank the reviewers for their comprehensive reviews and valuable feedback.  We are pleased that the reviewers recognized our work's innovative combination of heterogeneous graph representations and $SE(3)$-equivariant policies, noting its potential for broader generalization. The positive feedback on our thorough and statistically robust evaluations, as well as our insightful theoretical and empirical analysis, is greatly appreciated. Additionally, we are glad the clarity of our presentation and the detailed appendix were highlighted to enhance reproducibility.

A common theme among the reviews is questions about our experimental evaluation and concerns about the algorithm's applicability to a wide range of scenarios. To accommodate these issues, we conducted several additional experiments, which we present in the revision. To summarize:

- **New Task**: We introduced a new Rigid-Pushing task, as suggested by Reviewer **6nhr.** This task involves a rod pushing objects to a target position and orientation without physical attachment and provides a challenging testbed for continuous interaction dynamics. As shown in the revised Figure 3, HEPi performs better than EMPN and Transformer baselines, with faster convergence and higher returns.  A video showing example trajectories of this new task is also attached in the `Supplementary Material` revision.
- **Scalability to High-Resolution Objects:** To address Reviewer **qL9N**’s concern about scalability, we evaluated HEPi on high-resolution objects (average $> 1000$ nodes) in the *Rigid-Pushing* task. Without retraining, a HEPi agent trained on $< 30$ nodes (exact numbers vary between the objects, see Table 1 in the Appendix for full details), effectively scales to these higher-resolution inputs, enabled by the GNN's ability to exploit local structural patterns. This scalability is demonstrated in Figure 5, where HEPi consistently achieves high returns across resolutions.
- **Noise Sensitivity**: In line with concerns from Reviewers **qL9N** and **Z2z8** regarding HEPi’s applicability to real-world scenarios, we analyzed its robustness to noisy sensory inputs. While time constraints limited us from building a full vision pipeline, we opted for a systematic analysis by introducing Gaussian noise to simulate sensor inaccuracies. As shown in Figure 5, HEPi maintains strong performance at large noise levels, with only mild degradation under extreme noise, showcasing its robustness.

Additionally, we revised the manuscript in several places, fixed the mentioned typos, and tried to resolve the remaining unclear points. In our revision, these changes are marked in Blue.

Again, we thank the reviewers for their efforts and provide answers to their individual reviews to clarify and address their individual concerns.

---

> ### Author Response · Authors · 2024-11-28
>
> Dear All Reviewers,
>
> We hope that our additional empirical analyses and clarifications have satisfactorily addressed your concerns. As the discussion period deadline is coming closer, if you have any further questions, suggestions, or requests for additional explanations, we are happy to address them to the best of our ability.
>
> We deeply appreciate your engagement in this discussion, as your feedback is invaluable in helping us improve our paper.

---

### Meta-Review · Area_Chair_KM7V · 2024-12-24

**Metareview:**

The paper introduces Heterogeneous Equivariant Policy (HEPi), a graph-based policy model utilizing equivariant message passing networks to exploit geometric symmetries and explicitly model heterogeneity, enabling effective manipulation of rigid and deformable objects with multiple actuators, and demonstrating superior performance, sample efficiency, and generalization in a novel reinforcement learning benchmark.

All reviewers acknowledge the contributions of this work, emphasizing its (1) novelty, (2) potential for broader generalization, (3) thorough and statistically robust evaluations, (4) comprehensive empirical analysis, and (5) clear presentation.

During the Author-Reviewer Discussion phase, the authors provided thorough responses that successfully convinced some reviewers to raise their scores. All reviewers are in unanimous agreement to accept this paper. Still, the AC recommends that the authors carefully revisit both the original and post-rebuttal reviewer comments to ensure all concerns are adequately addressed in a revised version of the paper.

**Additional Comments On Reviewer Discussion:**

Since the reviewers were in unanimous agreement to accept this paper, no significant discussion took place during the Reviewer Discussion phase.

---

### Decision · Program_Chairs · 2025-01-22

Accept (Oral)